# Learning Optimal Tax Design in Nonatomic Congestion Games

**Qiwen Cui**
Paul G. Allen School of Computer Science
Engineering
University of Washington
Seattle, WA 98195
qwcui@cs.washington.edu

**Maryam Fazel**
Department of Electrical
Computer Engineering
University of Washington
Seattle, WA 98195
mfazel@uw.edu

**Simon S. Du**
Paul G. Allen School of Computer Science
Engineering
University of Washington
Seattle, WA 98195
ssdu@cs.washington.edu

## Abstract

In multiplayer games, self-interested behavior among the players can harm the social welfare. Tax mechanisms are a common method to alleviate this issue and induce socially optimal behavior. In this work, we take the initial step of learning the optimal tax that can maximize social welfare with limited feedback in congestion games. We propose a new type of feedback named *equilibrium feedback*, where the tax designer can only observe the Nash equilibrium after deploying a tax plan. Existing algorithms are not applicable due to the exponentially large tax function space, nonexistence of the gradient, and nonconvexity of the objective. To tackle these challenges, we design a computationally efficient algorithm that leverages several novel components: (1) a piece-wise linear tax to approximate the optimal tax; (2) extra linear terms to guarantee a strongly convex potential function; (3) an efficient subroutine to find the exploratory tax that can provide critical information about the game. The algorithm can find an $\epsilon$-optimal tax with $O(\beta F^2/\epsilon)$ sample complexity, where $\beta$ is the smoothness of the cost function and $F$ is the number of facilities.

## 1 Introduction

In modern society, large-scale systems often consist of many self-interested players with shared resources, such as transportation and communication networks. Importantly, the objectives of individual players are not always aligned with the system efficiency, and the system designer should take this into consideration. A widely known example is Braess's paradox, where adding more roads to a network can make the network more congested [Braess, 1968]. Price of anarchy is a notion that measures the inefficiency caused by selfish behavior compared with optimal centralized behavior [Koutsoupias and Papadimitriou, 1999]. Characterizing such inefficiency has been an active research area with applications in resource allocation [Marden and Roughgarden, 2014], traffic congestion [Roughgarden and Tardos, 2004], and others. The inefficiency motivates research on how to design mechanisms to improve performance even when the players are still behaving selfishly.

38th Conference on Neural Information Processing Systems (NeurIPS 2024).

Tax mechanisms are a standard approach to resolving the inefficiency issue, which are widely studied in economics, operations research, and game theory. The goal of tax mechanisms is to incentivize self-interested players to follow socially optimal behavior by applying tax/subsidy. Congestion game is a widely studied class of game theory models characterizing the interactions between players sharing facilities, where the cost of each facility depends on the "congestion" level [Wardrop, 1952, Rosenthal, 1973]. As a motivating example, in traffic routing games, each facility corresponds to an edge in a network, and each player chooses a path that connects her source node and target node. The cost of each facility corresponds to the latency of each edge, which depends on the number of players using that edge. Then, the tax can be interpreted as the toll collected by the road owner or the government to improve overall traffic efficiency [Bergendorff et al., 1997].

Most existing works on congestion game tax design focus on the computation complexity of the optimal tax [Nisan et al., 2007, Caragiannis et al., 2010]. They assume the tax designer has full knowledge of the underlying game, which is unrealistic in many applications. As Nash equilibrium is the only stable state of the system, we study a partial information feedback setting named "equilibrium feedback", where the tax designer can only observe information about the Nash equilibrium. The limited feedback information brings new challenges to the tax designer, and strategic exploration is necessary to learn or design the optimal tax. In this work, we aim to take the first step in learning optimal tax design for congestion games, and we study the following problem:

*How can we learn the optimal tax design in congestion games with equilibrium feedback?*

Below we highlight our contributions.

## 1.1 Main Contributions and Technical Novelties

**1. The first algorithm for learning optimal tax design in congestion games.** To the best of our knowledge, this is the first result for learning optimal tax in congestion games with partial information feedback. Our algorithm enjoys $O(F^2\beta/\epsilon)$ sample complexity for learning an $\epsilon$-optimal tax, where $F$ is the number of facilities and $\beta$ is the smoothness coefficient of the cost function. The sample complexity has no dependence on the number of actions, which could be exponential in $F$. In addition, we provide an efficient implementation for network congestion games with $\widetilde{O}(\text{poly}(V, E, \epsilon))$ computational complexity, where $V$ and $E$ are the numbers of the vertexes and edges in the network. Due to space limitation, we defer the computation analysis and experiments to Appendix C and Appendix E.

**2. Piece-wise linear function approximation.** We only assume the cost functions are smooth and make no parameterization assumptions as they are too strong to be satisfied in real-world applications. To tackle this challenge, we use piece-wise linear functions to approximate the optimal tax function. While only the values of the cost functions can be observed, we show that a carefully designed piece-wise linear function can approximate the unobservable optimal tax function well.

**3. Strongly convex potential function.** One challenge in tax design is controlling the sensitivity of Nash equilibrium w.r.t. tax perturbation. We always enforce tax functions with subgradient lower bounded by some positive value, which leads to a strongly convex potential function. As a result, the Nash equilibrium will be unique and Lipschitz with respect to tax perturbation. As the potential function for optimal tax is not necessarily strongly convex, we carefully choose the strong-convexity coefficient to balance the induced bias.

**4. Exploratory tax design.** Given the equilibrium feedback, the tax designer can only indirectly query the cost function by applying tax. Consequently, exploration in tax design becomes much more difficult than that in standard bandit problems where the player can directly query the value of an action [Lattimore and Szepesvári, 2020]. We design an exploratory tax that pushes the equilibrium to the "boundary", where an additional tax perturbation will change the equilibrium and reveal information about at least one unknown facility.

In this work, we focus on the well-known nonatomic congestion games. We hope our algorithm and analysis provide new insight on the intriguing structure of nonatomic congestion games. In addition, the tax design algorithm might find applications in real-world problems such as toll design in traffic networks. Due to space limitation, most proofs are deferred to the appendix.

**Notations.** $[m] = \{1, 2, \cdots, m\}$. For a set of real numbers $\mathcal{K}$ and a real number $x : \min\{\mathcal{K}\} \leq x \leq \max\{\mathcal{K}\}$, we define $[x]_{\mathcal{K}}^+ := \min_{y \in \mathcal{K}: y \geq x} y$ and $[x]_{\mathcal{K}}^- := \max_{y \in \mathcal{K}: y \leq x} y$. The clip operation $\mathrm{clip}(a, l, r) := \min\{\max\{a, l\}, r\}$ clips $a$ into the interval $[l, r]$. We use $O(\cdot)$ to hide absolute constants and $\widetilde{O}(\cdot)$ to hide polylog terms as well. A function $f : \mathcal{X} \mapsto \mathbb{R}$ is $\alpha$-strongly convex if $f(y) \geq f(x) + \nabla f(x)^\top (y - x) + \frac{\alpha}{2} \|y - x\|_2^2, \forall x, y \in \mathcal{X}$. $f$ is $\beta$-smooth if $\|\nabla f(x) - \nabla f(y)\|_2 \leq \beta \|x - y\|_2, \forall x, y \in \mathcal{X}$.

## 2 Related Work

**Learning in congestion games.** We refer the readers to the textbook [Nisan et al., 2007] for a general introduction to congestion games, the price of anarchy and tax mechanisms. Nonatomic congestion games were first studied in [Pigou, 1912] and formalized by [Wardrop, 1952]. Atomic congestion games were introduced by [Rosenthal, 1973] and the connection with potential games is developed by [Monderer and Shapley, 1996]. In contrast to general-sum games without structures, (approximate) Nash equilibrium can be computed efficiently in congestion games due to the existence of the potential function. Recently, various algorithms are developed to learn the Nash equilibrium in congestion games with different feedback oracles [Krichene et al., 2015, Chen and Lu, 2016, Cui et al., 2022, Jiang et al., 2022, Panageas et al., 2023, Dong et al., 2023, Dadi et al., 2024]. These algorithms are derived from the perspective of the players in the system, while our algorithm is essentially different in that it is utilized by the system designer to induce better equilibrium.

**Optimal tax design in congestion games.** For nonatomic congestion games, optimal tax design has a closed-form solution known as the marginal cost mechanism [Nisan et al., 2007]. For atomic congestion games, the marginal cost mechanism can no longer improve the efficiency [Paccagnan et al., 2021]. Instead, other mechanisms are proposed for optimal local/global and congestion dependent/independent tax in atomic congestion games [Caragiannis et al., 2010, Bilò and Vinci, 2019, Paccagnan et al., 2021, Paccagnan and Gairing, 2021, Harks et al., 2015]. Notably, all of these mechanisms assume full knowledge of the game while we consider learning with partial information feedback.

**Stackelberg games.** Stackelberg game [Von Stackelberg, 2010] models the interactions between leaders and followers such that leaders take actions first and the followers make decisions after observing leaders' actions. Tax design can be formulated as a Stackelberg game where the designer is the leader and the game players are the followers. Equipped with a best response oracle to predict followers' actions, Letchford et al. [2009], Blum et al. [2014], Peng et al. [2019] propose algorithms for learning Stackelberg equilibrium. Recently, Bai et al. [2021], Zhong et al. [2021], Zhao et al. [2023] generalize these results to learning Skackelberg equilibrium with bandit feedback, under finite actions or linear function approximation assumptions. For tax design, the search space is an exponentially large function space with complicated dependence on the objective. Consequently, existing results for Stackelberg games become vacuous when specialized to our problem.

**Mathematical programming under equilibrium constraint.** Tax design can be formulated as minimizing social cost with respect to tax under the constraint that players are following the equilibrium. This is known as mathematical programs with equilibrium constraints (MPEC). MPEC is a bilevel optimization problem and is NP-hard in general [Luo et al., 1996]. Existing approaches use specific inner loop algorithms to approach the equilibrium so that the gradient can be propagated to the outer loop [Li et al., 2020, Liu et al., 2022, Li et al., 2022, Maheshwari et al., 2023, Li et al., 2023, Grontas et al., 2024], relying on a unique and differentiable equilibrium [Colson et al., 2007]. However, such an approach requires many strong assumptions, such as the tax designer can control the algorithm of the agents, convex objective function and parameterized tax function. In contrast, our results make none of these assumptions.

## 3 Preliminaries

**Nonatomic congestion games.** A weighted nonatomic congestion game (congestion game) is described by the tuple $(\mathcal{F}, \mathcal{A}_{[m]}, w_{[m]}, c_{\mathcal{F}})$, where $\mathcal{F}$ is the set of facilities with cardinality $F$, $m$ is the number of commodities, $\mathcal{A}_i$ is the action set for commodity $i \in [m]$, $w_i \in [0, 1]$ is the weight for

commodity $i \in [m]$ such that $\sum_{i \in [m]} w_i = 1$, and $c_f : [0, 1] \mapsto [0, 1]$ is the cost function for facility $f \in \mathcal{F}$. Each commodity consists of infinite number of infinitesimal players with a total load to be $w_i$. Each individual player is self-interested and has a negligible effect on the game.

In congestion games, action $a \in \mathcal{A}_i, i \in [m]$ is a subset of $\mathcal{F}$, i.e. $a \subseteq \mathcal{F}$, which denotes the facilities utilized by action $a$. For commodity $i \in [m]$, we use strategy $x_i = (x_{i,a})_{a \in \mathcal{A}_i} \in [0, w_i]^{|\mathcal{A}_i|}$ with constraint $\sum_{a \in \mathcal{A}_i} x_{i,a} = w_i$ to denote how the load is distributed over all the actions. The joint strategy for the game is represented by $x = (x_1, x_2, \cdots, x_m) \in [0, 1]^A$, where $A = \sum_{i \in [m]} |\mathcal{A}_i|$. We use $\mathcal{X}$ to denote the set of all feasible strategies.

A decentralized perspective of strategy $x_i$ for commodity $i$ is that each self-interested infinitesimal player follows a randomized strategy that chooses $a \in \mathcal{A}_i$ with probability proportional to $x_{i,a}$. With the law of large number, the load on action $a$ would be $x_{i,a}$.

**Cost function.** For a strategy $x$, the cost of a facility is $c_f(l_f(x))$, where $l_f(x) = \sum_{i \in [m]} \sum_{a \in \mathcal{A}_i : f \in a} x_{i,a}$ is the load on facility $f$. The cost of an action $a$ is the sum of the facility cost that $a$ utilizes: $c_a(x) := \sum_{f \in a} c_f(l_f(x))$.

We make the following assumption on the cost function. Monotonicity is a standard congestion game assumption, which is also observed in many real-world applications as more players sharing one facility, each player will have less gain or more cost [Nisan et al., 2007]. Smoothness is a standard technical assumption for analysis.

**Assumption 1.** *We assume the cost function satisfies:*

1. *Monotonicity: $c_f(\cdot)$ is non-decreasing for all $f \in \mathcal{F}$,*

2. *Smoothness: $c_f(\cdot)$ is $\beta$-smooth for all $f \in \mathcal{F}$.*

**Nash equilibrium.** Nash equilibrium in nonatomic congestion games, also known as the Wardrop equilibrium [Wardrop, 1952], is the strategy that no player has the incentive to deviate from its strategy as formalized in Definition 1. In other words, Nash equilibrium is a stable state for a system with selfish players.

**Definition 1.** *A Nash equilibrium strategy $x$ is a joint strategy such that each player is choosing the best action: for any commodity $i \in [m]$ and actions $a, a' \in \mathcal{A}_i$, we have*

$$c_a(x) \leq c_{a'}(x), \text{ if } x_{i,a} > 0.$$

*Similarly, an $\epsilon$-approximate Nash equilibrium $x$ satisfies that*

$$\forall i \in [m], a, a' \in \mathcal{A}_i, c_a(x) \leq c_{a'}(x) + \epsilon, \text{ if } x_{i,a} > 0.$$

For a strategy $x$ and commodity $i$, actions $a \in \mathcal{A}_i$ such that $x_{i,a} > 0$ are named as the "in-support" actions and the others are "off-support" actions. For a Nash equilibrium, in-support actions must all have the same cost and off-support actions are no better than in-support actions. It is well known that Nash equilibrium always exists in congestion games [Beckmann et al., 1956].

**Potential Function.** An important concept in congestion games is the potential function:

$$\Phi(x) := \sum_f \int_0^{l_f(x)} c_f(u) du.$$

If Assumption 1 is satisfied, then $\Phi(x)$ is a convex function and Nash equilibrium is equivalent to the minimizer of the potential function [Beckmann et al., 1956].

**Network congestion games.** Network congestion games are congestion games with multicommodity network structure, which are also known as the selfish routing games [Roughgarden, 2005]. A multicommodity network is described by a directed graph $(\mathcal{V}, \mathcal{E})$ where $\mathcal{V}$ is the vertex set and $\mathcal{E}$ is the edge (facility) set. In addition, each commodity $i \in [m]$ corresponds to a pair of source and target vertex $(s_i, t_i)$, and actions are all feasible paths connecting $s_i$ and $t_i$. Each edge is associated with a nondecreasing cost (latency) function.

# 4  Tax Design for Congestion Games

In this section, we introduce tax design in congestion games. Before we get into the details, we will first introduce some notions to simplify the problem.

## 4.1  Polytope Description for Congestion Games

For a strategy $x_i \in \mathbb{R}^{|\mathcal{A}_i|}$, the dimension $|\mathcal{A}_i|$ can be as large as $2^F$. Instead, it would be convenient to consider the facility load $y_i \in \mathbb{R}^F$ such that $y_{i,f} = \sum_{a \in \mathcal{A}_i : f \in a} x_{i,a}$. In addition, we define $y = \sum_{i \in [m]} y_i \in \mathbb{R}^F$ to be the total facility load. We use $\phi_{(i)}(\cdot)$ to denote the reparameterization mapping:

$$\phi(x) = y, \phi_i(x_i) = y_i, \forall i \in [m],$$

and we set $\mathcal{Y} = \{y \in \mathbb{R}^F : \exists x \in \mathcal{X}, y = \phi(x)\}$ to be the set of all feasible loads. Note that $\phi$ is not necessarily a bijection, i.e., there could exist multiple strategies sharing the same load. We use $\phi^{-1}(y) := \{x \in \mathcal{X} : \phi(x) = y\}$ to denote the set of strategies that are mapped to load $y$. The potential function can be defined after the reparameterization as well:

$$\Phi^{\text{repa}}(y) := \sum_f \int_0^{y_f} c_f(u) du = \Phi(x), \forall x \in \Phi^{-1}(y).$$

Importantly, $\Phi^{\text{repa}}(y)$ does not depend on the choice of strategy $x \in \phi^{-1}(y)$. For the reparameterized potential function, we have the following lemma showing that it is almost equivalent to the original potential function. When it is clear from the context, we will simplify $\Phi^{\text{repa}}$ as $\Phi$.

**Lemma 1.** $\Phi^{\text{repa}}$ *is convex under Assumption 1. If* $y^* = \operatorname{argmin}_y \Phi^{\text{repa}}(y)$, *then for any* $x \in \phi^{-1}(y^*)$, *x is a Nash equilibrium.*

For any Nash equilibrium strategy $x$, we call $y = \phi(x)$ the Nash equilibrium load (Nash load).

## 4.2  Optimal Tax for Congestion Games

Nash equilibrium is a stable state for a system with self-interested players, as no player has the incentive to deviate unilaterally. However, Nash equilibrium does not efficiently utilize the facilities, which is measured by the social cost:

$$\Psi(y) := \sum_f y_f c_f(y_f).$$

Price of anarchy is a concept that measures the efficiency of selfish agents in a system, defined as the ratio between the worst-case social cost for equilibria and the optimal social cost:

$$\text{PoA} = \frac{\max_{y \text{ is a Nash equilibrium load}} \Psi(y)}{\min_{y \in \mathcal{Y}} \Psi(y)}$$

For example, in nonatomic congestion games with polynomial cost functions, the price of anarchy grows as $\Theta(d/\ln d)$ where $d$ is the degree of the polynomials [Nisan et al., 2007].

To reduce the price of anarchy, one standard approach is to enforce a tax on each facility to change the behavior of the self-interested players. Formally, a taxed congestion game is described by $(\mathcal{F}, \mathcal{A}_{[m]}, w_{[m]}, c_{\mathcal{F}}, \tau_{\mathcal{F}})$ with an additional tax function $\tau_f : [0, 1] \to \mathbb{R}$ on facility $f \in \mathcal{F}$. The cost of facility $f$ with load $u$ under tax becomes $c_f(u) + \tau_f(u)$. Correspondingly, we define the potential function with tax $\tau$ as

$$\Phi(y; \tau) := \sum_f \int_0^{y_f} [c_f(u) + \tau_f(u)] du,$$

and the Nash load would satisfy $y^* \in \operatorname{argmin}_y \Phi(y; \tau)$.

The optimal tax is defined as the tax that can induce optimal social behavior for self-interested players. We want to note that tax is not included in social cost following the convention in tax design.

---
**Protocol 1** Online Tax Design for Congestion Games
---
    **Initialize**: Facility set $\mathcal{F}$.
    **for** $t = 1, 2, \ldots, T$ **do**
        Designer chooses tax $\tau^t$.
        Designer observes Nash load $y^t = \operatorname{argmin}_{y \in \mathcal{Y}} \Phi(y; \tau)$ and Nash cost $c^t = [c_f(y_f^t)]_{f \in \mathcal{F}}$ for
    $f \in \mathcal{F}$.
    **end for**
---

**Definition 2.** *A tax $\tau$ is an optimal tax if all Nash equilibria under tax $\tau$ can minimize the social cost:*

$$\operatorname*{argmin}_{y \in \mathcal{Y}} \Phi(y; \tau) \subseteq \operatorname*{argmin}_{y \in \mathcal{Y}} \Psi(y).$$

*In addition, a tax $\tau$ is an $\epsilon$-optimal tax if we have*

$$\Psi(y) \leq \min_{y' \in \mathcal{Y}} \Psi(y') + \epsilon, \forall y \in \operatorname*{argmin}_{y'' \in \mathcal{Y}} \Phi(y''; \tau).$$

The marginal cost tax is defined as

$$\tau^* : \tau_f^*(u) = uc_f'(u), \forall f \in \mathcal{F}.$$

As $\Phi(y; \tau^*) = \Psi(y)$, the Nash equilibrium under tax $\tau^*$ will minimize the social cost and $\tau^*$ is an optimal tax [Nisan et al., 2007]. We will make the following assumption so that the cost combined with tax $c + \tau^*$ is still non-decreasing. In many real world problems, $c_f'(u)$ is non-decreasing due to the law of diminishing marginal utility, which guarantees Assumption 2.

**Assumption 2.** *Marginal cost tax $\tau_f^*(u) = uc_f'(u)$ is non-decreasing for all $f \in \mathcal{F}$.*

### 4.3 Tax Design for Congestion Games

In this paper, we consider the case where the system designer (e.g. government) wants to enforce an (approximate) optimal tax to induce optimal social behavior and maximize social welfare. However, the cost function is unknown so the optimal tax function cannot be computed directly via the marginal cost mechanism. On the other hand, the designer can enforce several taxes and observe the feedback. As Nash equilibrium is the stable state of the system, we assume the designer can observe the equilibrium feedback.

Formally, tax design is formulated as an online learning problem as shown in Protocol 1. At round $t$, the designer can choose a tax $\tau^t$ and observe the corresponding Nash equilibrium load $y^t \in \mathbb{R}^F$ and Nash equilibrium cost $c^t \in \mathbb{R}^F$. The sample complexity of a tax design algorithm is the number of rounds for designing an $\epsilon$-optimal tax.

A naive approach is the designer first enumerates all of the $\epsilon$-approximations of $\tau^*$ and chooses the tax with minimal social cost. However, such an approach would require $O((1/\epsilon)^{F\beta/\epsilon})$ samples as the complexity of using piece-wise linear function to approximate $\tau_f^*$ (a $\beta$-smooth function) with $\epsilon$ error is $O((1/\epsilon)^{\beta/\epsilon})$, resulting in exponential dependence on the parameters $\beta$, $1/\epsilon$ and $F$.

Another approach is applying algorithms for mathematical programming under equilibrium constraints. Specifically, we can formulate tax design as solving

$$\min_{\tau} \Psi(y(\tau)), \text{ s.t. } y(\tau) = \operatorname*{argmin}_{y \in \mathcal{Y}} \Phi(y; \tau).$$

However, $y(\tau)$ can be non-differentiable or even discontinuous w.r.t. $\tau$, and $\Psi(y(\tau))$ can be non-convex w.r.t. $\tau$ (Lemma 5). As a result, previous results do not apply to our problem as they apply gradient-based methods and make convexity assumptions [Li et al., 2020, Liu et al., 2022].

## 5 Learning Optimal Tax in Nonatomic Congestion Games

In this section, we describe our algorithm that can learn an $\epsilon$-optimal tax with $O(F^2\beta/\epsilon)$ samples. First, we introduce piece-wise linear functions as a nonparametric way to approximate the marginal cost tax $\tau^*$ [Takezawa, 2005].

**Definition 3.** *(Piece-wise Linear Function) We use a dictionary\* $d = \{(x_1, y_1), \cdots, (x_n, y_n)\}$ for $x_i \neq x_j, \forall i \neq j$ (w.l.o.g. we let $x_1 < x_2 < \cdots < x_n$) to represent a piece-wise linear function $d(\cdot)$ on $[x_1, x_n]$ such that*

$$d(x) = \frac{x - x_{i+1}}{x_i - x_{i+1}} y_i + \frac{x_i - x}{x_i - x_{i+1}} y_{i+1}, \forall x \in [x_i, x_{i+1}].$$

*In addition, we use $\bigcup$ to represent the update method for dictionary. I.e., $d\bigcup(x, y)$ is the piece-wise linear function interpolating one more point $(x, y)$ if $(x, d(x))$ is not already in $d$, otherwise it will update $d(x)$ to $y$.*

We will maintain the piece-wise function on a grid $\mathcal{L} = \{0, \Delta, 2\Delta, \cdots, K\Delta = 1\}$ with $K = \left\lceil \frac{2\beta}{\epsilon} \right\rceil$ and $\Delta = 1/K$. The time complexity for computing $d(x)$ is $O(\log K)$ for any $x \in [0, 1]$.

## 5.1 Main Algorithm

---
**Algorithm 1** Tax Design for Congestion Game
---
1: **Initialize**: Facility set $\mathcal{F}$, number of rounds $T$, tolerance $\epsilon$, smoothness $\beta$, perturbation $\delta = \epsilon \Delta^2 / 8$.
2: Set initial tax $\tau^1 : \tau_f^1 = \{(0, 0), (1, \beta + \epsilon)\}$ for all $f \in \mathcal{F}$. Set $\mathcal{K}_f^1$ to be $\{0\}$ for all $f \in \mathcal{F}$.
3: **for** $t = 1, 2, \ldots, T$ **do**
4:      Observe Nash load $y^t \in \mathbb{R}^F$ and Nash cost $c^t \in \mathbb{R}^F$ under tax $\tau^t$.
5:      Set $\bar{\mathcal{F}}$ to be the unknown facility set (Definition 4).
6:      Set $l_f = \tau_f^t([y_f^t]_{\mathcal{K}_f^t}^-) + \epsilon(y_f^t - [y_f^t]_{\mathcal{K}_f^t}^-)$ and $r_f = \tau_f^t([y_f^t]_{\mathcal{K}_f^t \bigcup \{1\}}^+) + \epsilon(y_f^t - [y_f^t]_{\mathcal{K}_f^t \bigcup \{1\}}^+)$ for
     each $f \in \bar{\mathcal{F}}$.
7:      Run Algorithm 2 with input $y^t, c^t, \tau^t = [\tau_f^t(y_f^t)]_f, \bar{\mathcal{F}}$ and $[l_f, r_f]_{f \in \bar{\mathcal{F}}}$.
8:      **if** Algorithm 2 return False **then**
9:          **return** $\tau^t$
10:      **else**
11:          Algorithm 2 return $\widetilde{\tau} \in \mathbb{R}^F, \widetilde{f} \in \bar{\mathcal{F}}, \text{sign} \in \{-1, 1\}$.
12:          Apply tax $\dot{\tau}^t : \dot{\tau}_{\widetilde{f}}^t = \tau_{\widetilde{f}}^t \bigcup(y_{\widetilde{f}}^t, \widetilde{\tau}_{\widetilde{f}}^t) + \text{sign} \cdot \delta$ and $\dot{\tau}_f^t = \tau_f^t \bigcup(y_f^t, \widetilde{\tau}_f^t)$ for $f \neq \widetilde{f}$.
13:          Observe $\dot{y}^t, \dot{c}^t \in \mathbb{R}^F$ as the Nash load and the Nash cost of each facility.
14:          Update $\tau_{t+1}$ and $\mathcal{K}_{t+1}$ according to (1).
15:      **end if**
16: **end for**
---

In this section, we introduce our main algorithm. At each round $t$, we will maintain a known index set $\mathcal{K}_f^t \subseteq \mathcal{L}$ where the marginal cost tax can be accurately estimated (Lemma 7), and use a piece-wise linear function to approximate the tax function by interpolating the values at the known indexes. The piece-wise linear function takes the form $\tau_f^t = \{(x_i^t, y_i^t)\}_i$ and the known index set $\mathcal{K}_f^t$ satisfies $\{x_i^t\}_i = \mathcal{K}_f^t \bigcup \{1\}$ and $\mathcal{K}_f^t \subseteq \mathcal{L}$. Here 1 is a special case as it is not in the known index set initially but it is needed as the boundary for the piece-wise linear function $\tau_f^t$. Initially, the tax is set to be $\tau_f^1(u) = \{(0, 0), (1, \beta + \epsilon)\}$ and the auxiliary tax is $\widehat{\tau}_f^1 = \{(0, 0), (1, \beta)\}$ for $f \in \mathcal{F}$ (Line 2). Here we set $\widehat{\tau}_f^1(1) = \beta$ as $\beta$ is always an upper bound on $\tau_f^*(1)$. The auxiliary tax $\widehat{\tau}_f^t$ is a non-decreasing piece-wice linear approximation of $\tau_f^*$ and we always set tax $\tau_f^t(u) = \widehat{\tau}_f^t(u) + \epsilon u$ to ensure that the subgradient of the tax enforced is lower bounded by $\epsilon$.

At round $t$, after observing Nash equilibrium load $y^t \in \mathbb{R}^F$ and Nash equilibrium cost $c^t \in \mathbb{R}^F$, the facilities are split into two sets: known facilities and unknown facilities.

**Definition 4.** *For each round $t$, facility $f$ is known if the Nash load $y_f^t \in [0, 1]$ satisfies $[y_f^t]_{\mathcal{L}}^- \in \mathcal{K}_f^t$ and $[y_f^t]_{\mathcal{L}}^+ \in \mathcal{K}_f^t$. Otherwise, facility $f$ is unknown for round $t$.*

For a known facility $f$, the Nash load is either in the known index set or sandwiched by two consecutive known indexes. As a result, the tax estimate for the Nash load $\tau_f^t(y_f^t)$ will be close to the

---
    \*In this dictionary, key is $x_i$ and value is $y_i$. For readers unfamiliar with the dictionary data structure, it can be regarded as a set with a special update operation.

true optimal tax $\tau_f^*(y_f^t)$ with error $2\epsilon$ (Lemma 8). We will apply Algorithm 2 to find the exploratory tax to gather information about unknown facilities (Line 7).

**Proposition 1.** *If Algorithm 2 return* False *at round t, then tax $\tau^t$ is an $6\epsilon F$-optimal tax. If Algorithm 2 output $\widetilde{\tau}^t, \widetilde{f}^t, \mathrm{sign}^t$ at round t, then we have*

$$0 < \left| y_{\widetilde{f}^t}^t - \dot{y}_{\widetilde{f}^t}^t \right| \le \Delta.$$

If Algorithm 2 output $\widetilde{\tau}^t, \widetilde{f}^t, \mathrm{sign}^t$ at round $t$, we update the tax and the known index set by the following rule. For $u \in \{[y_{\widetilde{f}^t}^t]_{\mathcal{L}}^+, [y_{\widetilde{f}^t}^t]_{\mathcal{L}}^-\} \backslash \mathcal{K}_{\widetilde{f}^t}^t$ (this set is not empty as $\widetilde{f}^t$ is an unknown facility), we set

$$\widehat{\tau}_{\widetilde{f}^t}^{t+1} = \widehat{\tau}_{\widetilde{f}^t}^t \bigcup \left( u, \mathrm{clip}\left( u \cdot \frac{c_{\widetilde{f}^t}^t - \dot{c}_{\widetilde{f}^t}^t}{y_{\widetilde{f}^t}^t - \dot{y}_{\widetilde{f}^t}^t}, \widehat{\tau}_{\widetilde{f}^t}^t([y_{\widetilde{f}^t}^t]_{\mathcal{K}_{\widetilde{f}^t}^t}^-), \widehat{\tau}_{\widetilde{f}^t}^t([y_{\widetilde{f}^t}^t]_{\mathcal{K}_{\widetilde{f}^t}^t \bigcup \{1\}}^+) \right) \right), \qquad (1)$$

$$\mathcal{K}_{\widetilde{f}^t}^{t+1} = \mathcal{K}_{\widetilde{f}^t}^t \bigcup \{u\}. \qquad (2)$$

and $\widehat{\tau}_f^{t+1} = \widehat{\tau}_f^t, \mathcal{K}_f^{t+1} = \mathcal{K}_f^t$ for $f \ne \widetilde{f}^t$. Then we set $\tau_f^{t+1}(u) = \widehat{\tau}_f^{t+1}(u) + \epsilon u$ for all $f \in \mathcal{F}$ and $u \in [0, 1]$.

In words, we clip the two-point estimate $u \cdot \frac{c_{\widetilde{f}^t}^t - \dot{c}_{\widetilde{f}^t}^t}{y_{\widetilde{f}^t}^t - \dot{y}_{\widetilde{f}^t}^t}$ on the left and right known index of $\widehat{\tau}_{\widetilde{f}^t}^t(u)$ so that $\widehat{\tau}_{\widetilde{f}^t}^{t+1}(u)$ is still a non-decreasing piece-wise linear approximation of the marginal cost tax $\tau_f^*$. $\tau_f^{t+1}$ is added with an extra linear term to guarantee a strongly convex potential function (Lemma 2). As $0 < \left| y_f^t - \dot{y}_f^t \right| \le \Delta$, the two point estimate of the gradient $\frac{c_f^t - \dot{c}_f^t}{y_f^t - \dot{y}_f^t}$ is accurate enough for $c_f'(u)$ such that $\left| \tau_{\widetilde{f}^t}^{t+1}(u) - \tau_f(u) \right| \le \epsilon$ (Lemma 6).

As $|\mathcal{K}_{\widetilde{f}^t}^t|$ increases by 1 at round $t$ and there are $F$ such sets with size bounded by $O(\beta/\epsilon)$, Algorithm 2 will output False within at most $O(F\beta/\epsilon)$ rounds, which implies $\tau^t$ is an $\epsilon F$-optimal tax (Proposition 1). With proper rescaling, the sample complexity for learning $\epsilon$-optimal tax is $O(F^2\beta/\epsilon)$.

**Theorem 1.** *Under Assumption 1 and Assumption 2, Algorithm 1 will output a $6\epsilon F$ tax within $T \le 2F\beta/\epsilon$ rounds. In addition, each round has at most two tax realizations.*

**Remark 1.** *To uniformly approximate a $\beta$-smooth function, we have to know its value at $O(\beta/\epsilon)$ points [Takezawa, 2005]. For an $\epsilon$-optimal tax, we need to estimate $\tau_f^*$ with $\epsilon/F$ accuracy as the error accumulates with all the facilities. As a result, we conjecture that $O(F^2\beta/\epsilon)$ sample complexity is tight and we leave the lower bound to future work.*

**Remark 2.** *Our algorithm can be easily adapted to the case where we have feedback other than only the equilibrium feedback. Specifically, when the tax designer obtain a non-equilibrium feedback, she can still update the optimal tax estimate if the feedback provides new information. It is possible for our algorithm to find the optimal tax even if no equilibrium feedback is provided. In addition, as long as the equilibrium can be reached after applying a tax, the algorithm can always find the optimal tax.*

## 5.2 Subroutine for Finding Exploratory Tax

In this section, we describe Algorithm 2, which can find an exploratory tax that satisfies Proposition 1. The idea is we can observe another similar but different Nash equilibrium load by perturbing the tax. However, there are two challenges:

1. Perturbing the tax might change the Nash equilibrium load drastically.
2. Perturbing the tax might not change the Nash equilibrium load at all.

To resolve the first issue, we always apply taxes that have (sub)gradient lower bounded by $\epsilon > 0$. The feasible range $[l_f, r_f]$ for updating tax $\tau_f^t$ with $(y_f^t, \cdot)$ guarantees that the updated tax still maintains the subgradient lower bound. By Lemma 2, the potential function is always $\epsilon$-strongly convex. As a result, the Nash load for any feasible tax is unique and Lipschitz w.r.t. tax perturbation. To resolve

---

**Algorithm 2** Test Tax Design

---

1: **Initialize**: Nash flow $y \in \mathbb{R}^F$, tax $\tau \in \mathbb{R}^F$, cost $c \in \mathbb{R}^F$, unknown facility set $\bar{\mathcal{F}}$, unknown facility range $[l_f, r_f]$ for $f \in \bar{F}$.
2: Set strategy $x \in \phi^{-1}(y)$. Compute commodity load $y_i = \phi_i(x_i) \in \mathbb{R}^F$ for $i \in [m]$.
3: Set $I = $ False.
4: **if** Exists $f \in \bar{\mathcal{F}}$ and $i \in [m]$ such that $0 < y_i(f) < w_i$ **then**
5:     **return** $\tau, f, 1$.
6: **end if**
7: **for** Commodity $i \in [m]$ **do**
8:     Let $\bar{\mathcal{F}}_i = \{f \in \bar{\mathcal{F}} : \sum_{a:f\in a} x_{i,a} = w_i\}$ and $\bar{\mathcal{F}}_i' = \{f \in \bar{\mathcal{F}} : \sum_{a:f\in a} x_{i,a} = 0\}$.
9:     Set $\bar{\tau} : \bar{\tau}_{\bar{\mathcal{F}}_i} = r_{\bar{\mathcal{F}}_i}, \bar{\tau}_{\bar{\mathcal{F}}_i'} = l_{\bar{\mathcal{F}}_i'}, \bar{\tau}_{\mathcal{F}\setminus(\bar{\mathcal{F}}_i \bigcup \bar{\mathcal{F}}_i')} = \tau_{\mathcal{F}\setminus(\bar{F}_i \bigcup \bar{\mathcal{F}}_i')}$.
10:     **if** $\text{Gap}_i(x, c + \bar{\tau}) < 0$ **then**
11:         Set $I = $ True.
12:         **break**
13:     **end if**
14: **end for**
15: **if** $I = $ False **then**
16:     **return** False.
17: **end if**
18: Set $\tau' = \tau$
19: **for** $f \in \bar{\mathcal{F}}_i$ **do**
20:     Set $\tilde{\tau}^u : \tilde{\tau}_f^u = u, \tilde{\tau}_{\mathcal{F}\setminus\{f\}}^u = \tau'_{\mathcal{F}\setminus\{f\}}$.
21:     Set $u = \text{argmax}\{u : \text{Gap}_j(x, c + \tilde{\tau}^u) \geq 0, \forall j\}$.
22:     **if** $u \leq r_f$ **then**
23:         **return** $\tilde{\tau}^u, f, 1$.
24:     **end if**
25:     Set $\tau' = \tilde{\tau}^{r_f}$.
26: **end for**
27: **for** $f \in \bar{\mathcal{F}}_i'$ **do**
28:     Set $\tilde{\tau}^u : \tilde{\tau}_f^u = u, \tilde{\tau}_{\mathcal{F}\setminus\{f\}}^u = \tau'_{\mathcal{F}\setminus\{f\}}$.
29:     Set $u = \text{argmin}\{u : \text{Gap}_j(x, c + \tilde{\tau}^u) \geq 0, \forall j\}$.
30:     **if** $u \geq l_f$ **then**
31:         **return** $\tilde{\tau}^u, f, -1$.
32:     **end if**
33:     Set $\tau' = \tilde{\tau}^{l_f}$.
34: **end for**

---

the second issue, we find the tax that makes the current Nash equilibrium on the "boundary". I.e., an additional perturbation will make the Nash equilibrium change. Intuitively, this is similar to removing the slackness in a constrained optimization problem. By Lemma 3, we can observe a different Nash load on $f$ if we make the additional perturbation.

**Lemma 2.** *If the subgradient of the cost function $c_f$ is lower bounded by $\epsilon > 0$ for all $f \in \mathcal{F}$, then the potential function $\Phi^{\text{repa}}(y)$ is $\epsilon$-strongly convex. However, $\Phi(x)$ is not necessarily strongly convex.*

**Lemma 3.** *If two taxes $\tau$ and $\dot{\tau}$ only differ in facility $f$ and the Nash loads $y$ and $\dot{y}$ are different, then $y_f \neq \dot{y}_f$.*

**Definition 5.** *The gap for a strategy $x \in \mathcal{X}$ with cost $c \in \mathbb{R}^F$ is defined as*

$$\text{Gap}_i(x, c) = \min_{a:x_{i,a}=0} \sum_{f:f\in a} c_f - \max_{a:x_{i,a}\neq 0} \sum_{f:f\in a} c_f. \tag{3}$$

In the algorithm, we use $\text{Gap}_i(x, c)$ to measure the cost gap between in-support actions and off-support actions for commodity $i$ and strategy $x$. If $x$ is a Nash equilibrium and $c$ is the Nash cost, then all of the in-support actions have the same minimal cost and $\text{Gap}_i(x, c) \geq 0$ holds. Informally, "boundary" tax $\tau$ means that $\text{Gap}_i(x, c + \tau) = 0$ for a Nash equilibrium $x$ and perturbing $\tau$ results in $\text{Gap}_i(x, c + \tau) < 0$, so the Nash equilibrium under the perturbed tax will be different from $x$.

Now we discuss how Algorithm 2 finds the "boundary" tax in detail. The input to the algorithm is the Nash flow $y$, the Nash cost $c$, the Nash tax $\tau$, the unknown facility set $\bar{\mathcal{F}}$ and the feasible tax range $[l_f, r_f]$ for each unknown facility $f \in \bar{\mathcal{F}}$. We emphasize that here the Nash cost/tax are the values of the cost/tax function on the Nash load and they are vectors in $\mathbb{R}^F$ instead of functions. $[l_f, r_f]$ is the feasible range for the perturbed tax value at facility $f$. By the definition of $l_f$ and $r_f$, current tax $\tau_f^t$ updated with $(y_f^t, u), u \in [l_f, r_f]$ is still a tax with subgradient lower bounded by $\epsilon$.

For the first step, the algorithm will compute strategy $x \in \phi^{-1}(y)$ as the Nash equilibrium strategy (Line 2). If there exists an unknown facility $f$ and commodity $i$ such that not all load of commodity $i$ is using $f$ or not using $f$ (Line 4), then perturbing the tax at facility $f$ will make $x$ not longer a Nash equilibrium as in-support actions have different costs.

Otherwise, for each unknown facility $f$ and commodity $i$, either all of the load is using $f$ or all of the load is not using $f$. As a result, in-support actions always have the same cost after perturbing the tax. For the next step, we verify if there exists a tax within the feasible ranges for unknown facilities that makes $x$ not a Nash equilibrium. However, there does not exist a universal worst-case tax that can verify if $x$ is always a Nash equilibrium or not.

Fortunately, the worst-case tax has a closed form for each commodity separately: the taxes for facilities used by all of the Nash load would be the upper bound $r_f$ and the taxes for facilities used by none of the Nash load would be the lower bound $l_f$, thus maximizing the cost for in-support actions and minimizing the cost for off-support actions (Line 9). For each commodity $i$, we apply the corresponding worst-case tax and check if the in-support actions are still the optimal actions (Line 10). If for all commodities, the in-support actions are optimal under the worst-case tax, then for any tax within the feasible range, $y$ is the Nash load and the algorithm will output False (Line 16). As $\tau^*$ is approximately within the range, $y^t$ approximately minimizes the social cost (Lemma 10).

Otherwise, the algorithm finds commodity $i$ such that $x$ is not the Nash equilibrium under the worst-case tax (Line 11). For the last step, we gradually transform the initial tax to this worst-case tax and stop when $x$ is not the Nash equilibrium for some commodity. Specifically, the algorithm iteratively changes the tax in the unknown facility set $\bar{\mathcal{F}} = \bar{\mathcal{F}}_i \bigcup \bar{\mathcal{F}}_i'$ (Line 19 and Line 27) to the worst-case tax.

For facility $f \in \bar{\mathcal{F}}_i$, the algorithm finds the boundary tax for facility $f$ that satisfies

$$u = \operatorname*{argmax}_{u} \{u : \mathrm{Gap}_j(x, c + \widetilde{\tau}^u) \geq 0, \forall j \in [m]\}.$$

If $u \leq r_f$, we will output $\widetilde{\tau}^u, f, 1$. By the definition of $u$, if we further increase $u$, one of the gaps will become negative and $x$ is no longer the Nash equilibrium. Otherwise, all feasible taxes for $f$ have a nonnegative gap for all commodities, which means $x$ is still the Nash equilibrium, and we continue for the next facility. After enumerating all the facilities in $\bar{\mathcal{F}}_i$, we enumerate $\bar{\mathcal{F}}_i'$ in the same way. Eventually, the tax is transformed into the worst-case tax with negative gap for commodity $i$, so this process will end and output $\widetilde{\tau}^u, f, \mathrm{sign}$ such that $\widetilde{\tau}^u$ is the tax that makes the Nash equilibrium on the boundary, $f$ is the facility to perturb and $\mathrm{sign}$ is the direction to perturb the tax at $f$.

## 6 Conclusion

We proposed the first algorithm with polynomial sample complexity for learning optimal tax in nonatomic congestion games. The algorithm leverages several novel designs to exploit the special structure of congestion games, which can also be implemented efficiently. Below we list a few potential future research directions:

1. Relaxing the Nash equilibrium assumption to players following no-regret dynamics or quantal response equilibrium.
2. Design algorithms that do not require prior knowledge of the smoothess coefficient.
3. Generalize the algorithm to atomic congestion games.

## 7 Acknowledgement

SSD acknowledges the support of NSF IIS 2110170, NSF DMS 2134106, NSF CCF 2212261, NSF IIS 2143493, NSF CCF 2019844, and NSF IIS 2229881. MF acknowledges the support of NSF awards CCF 2007036, TRIPODS II DMS 2023166, CCF 2212261, and CCF-AF 2312775.

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

# A  Basics about Congestion Games

**Lemma 4.** *If strategy $x$ is an $\epsilon$-NE in a congestion game, then $x$ is an $\epsilon$-minimizer of the corresponding potential function $\Phi(\cdot)$.*

*Proof.* Let $x^* = \operatorname{argmin}_{x \in \mathcal{X}} \Phi(x)$ and $y = \phi(x)$. First, we show that

$$
\begin{aligned}
\nabla_{i,a} \Phi(x) &= \nabla_{i,a} \sum_f \int_0^{y_f} c_f(u)du \\
&= \nabla_{i,a} \sum_{f \in a} \int_0^{y_f} c_f(u)du \\
&= \sum_{f \in a} c_f(y_f) \nabla_{i,a} y_f \\
&= \sum_{f \in a} c_f(y_f) \nabla_{i,a} \sum_{i',a':f \in a'} x_{i',a'} \\
&= \sum_{f \in a} c_f(y_f).
\end{aligned}
$$

Then we have

$$
\begin{aligned}
\Phi(x) - \Phi(x^*) &\le \langle x - x^*, \nabla \Phi(x) \rangle && \text{(Convexity)} \\
&\le \sum_{i \in [m]} \langle x_i - x_i^*, \nabla_i \Phi(x) \rangle \\
&\le \sum_{i \in [m]} \left[ \sum_{a \in \mathcal{A}_i} x_{i,a} \sum_{f \in a} c_f(y_f) - \min_{a \in \mathcal{A}_i} w_i \sum_{f \in a} c_f(y_f) \right] \\
&\le \sum_{i \in [m]} \sum_{a \in \mathcal{A}_i} x_{i,a} \epsilon \\
&= \sum_{i \in [m]} w_i \epsilon \\
&= \epsilon.
\end{aligned}
$$

$\square$

**Lemma 1.** $\Phi^{\mathrm{repa}}$ *is convex under Assumption 1. If $y^* = \operatorname{argmin}_y \Phi^{\mathrm{repa}}(y)$, then for any $x \in \phi^{-1}(y^*)$, $x$ is a Nash equilibrium.*

*Proof.* For $y^1, y^2 \in \mathcal{Y}$, we have

$$
\Phi^{\mathrm{repa}}(y^1) + \Phi^{\mathrm{repa}}(y^2) - 2\Phi^{\mathrm{repa}}\left(\frac{y^1 + y^2}{2}\right) = \sum_f \left[ \int_0^{y_f^1} c_f(u)du + \int_0^{y_f^2} c_f(u)du - 2 \int_0^{\frac{y_f^1 + y_f^2}{2}} c_f(u)du \right].
$$

Now we show that $\int_0^{y_f^1} c_f(u)du + \int_0^{y_f^2} c_f(u)du - 2 \int_0^{\frac{y_f^1 + y_f^2}{2}} c_f(u)du$ is nonnegative for all $f \in \mathcal{F}$. W.l.o.g., we assume $y_f^1 \le y_f^2$ and we have

$$
\begin{aligned}
\int_0^{y_f^1} c_f(u)du + \int_0^{y_f^2} c_f(u)du - 2 \int_0^{\frac{y_f^1 + y_f^2}{2}} c_f(u)du &= \int_{\frac{y_f^1 + y_f^2}{2}}^{y_f^2} c_f(u)du - \int_{y_f^1}^{\frac{y_f^1 + y_f^2}{2}} c_f(u)du \\
&= \int_{y_f^1}^{\frac{y_f^1 + y_f^2}{2}} \left[ c_f(u + \frac{y_f^2 - y_f^1}{2}) - c_f(u) \right] du \\
&\ge 0,
\end{aligned}
$$

where the last step is from Assumption 1 (monotonicity). As a result, $\Phi^{\mathrm{repa}}$ is convex.

Let $y^* = \mathrm{argmin}_y\, \Phi^{\mathrm{repa}}(y)$ and $x \in \phi^{-1}(y^*)$. If there exists $x' \in \mathcal{X}$ such that $\Phi(x') < \Phi(x)$, then we have $\Phi^{\mathrm{repa}}(\phi(x')) < \Phi^{\mathrm{repa}}(y^*)$, which contradicts the definition of $y^*$. As a result, $x$ is the minimizer of $\Phi(\cdot)$, which means $x$ is a Nash equilibrium. $\qquad\square$

**Lemma 5.** *The Nash load under tax $\tau$: $y(\tau) = \mathrm{argmin}_{y \in \mathcal{Y}}\, \Phi(y;\tau)$ is not continuous w.r.t. $\tau$. In addition, the social welfare $\Psi(y(\tau))$ is not convex w.r.t. $\tau$.*

*Proof.* For the first part, we construct a congestion game with two facilities $f_1, f_2$, one commodity with action set $\{f_1, f_2\}$, and constant cost $c_1 = 1, c_2 = 1 - \epsilon$ with $\epsilon > 0$. Then for tax $\tau = 0$, we have $y(\tau) = [0, 1]$. For constant tax $\tau_1 = 0, \tau_2 = 2\epsilon$, we have $y(\tau) = [1, 0]$. As $\epsilon$ can be arbitrarily small, $y(\tau)$ is not continuous w.r.t. $\tau$.

For the second part, we construct a congestion game with two facilities $f_1, f_2$, one commodity with action set $\{f_1, f_2\}$, and cost function $c_1 = 1, c_2(u) = \sqrt{u}$ for $u \in [0, 1]$. We apply constant tax $\tau : \tau_1 = t, \tau_2 = 0$ for $t \in [-1, 0]$. The Nash equilibrium under tax $\tau$ is $y(\tau) = [1 - (1+t)^2, (1+t)^2]$. Then the social cost is $\Psi(y(\tau)) = 1 - t(1 + t)^2$, which is not convex on $[-1, 0]$. $\qquad\square$

# B  Missing Proofs in Section 5

**Lemma 2.** *If the subgradient of the cost function $c_f$ is lower bounded by $\epsilon > 0$ for all $f \in \mathcal{F}$, then the potential function $\Phi^{\mathrm{repa}}(y)$ is $\epsilon$-strongly convex. However, $\Phi(x)$ is not necessarily strongly convex.*

*Proof.* First, by the definition of the potential function $\Phi$, it is easy to show that $\nabla\Phi(y) = [c_f(y_f)]_{f \in \mathcal{F}}$. For $y^1, y^2 \in \mathcal{Y}$, we have

$$(\nabla\Phi(y^1) - \nabla\Phi(y^2))^\top (y^1 - y^2) = \sum_{f \in \mathcal{F}} (c_f(y_f^1) - c_f(y_f^2))(y_f^1 - y_f^2) \geq \sum_{f \in \mathcal{F}} \epsilon (y_f^1 - y_f^2)^2 = \epsilon \left\| y^1 - y^2 \right\|_2^2,$$

which implies $\Phi(\cdot)$ is a $\epsilon$-strongly convex function.

For the second argument, we only need to construct a congestion game such that there exists two strategy $x^1, x^2 \in \mathcal{X}$ such that for $\phi(tx^1 + (1-t)x^2)$ is a constant for $t \in [0, 1]$, which implies the potential function $\Phi(tx^1 + (1-t)x^2) = \Phi^{\mathrm{repa}}(\phi(tx^1 + (1-t)x^2))$ is a constant w.r.t. $t$. However, a strongly convex function cannot be a constant on a line, which implies $\Phi$ is not strongly convex.

We construct a congestion game with three facilities $f_1, f_2, f_3$, three actions $a_1 = \{f_1\}, a_2 = \{f_2\}, a_3 = \{f_3\}$ and three commodities with action set $\{a_1, a_2\}, \{a_2, a_3\}, \{a_3, a_1\}$. Strategy $x^1$ : $x_1^1 = [1, 0, 0], x_2^1 = [0, 1, 0], x_3^1 = [0, 0, 1]$ and $x^2$ : $x_1^2 = [0, 1, 0], x_2^2 = [0, 0, 1], x_3^2 = [1, 0, 0]$. Then $tx^1 + (1-t)x^2$ is a feasible strategy and we have $\phi(tx^1 + (1-t)x^2) = [1, 1, 1]$ for all $t \in [0, 1]$. $\qquad\square$

**Lemma 3.** *If two taxes $\tau$ and $\dot{\tau}$ only differ in facility $\dot{f}$ and the Nash loads $y$ and $\dot{y}$ are different, then $y_{\dot{f}} \neq \dot{y}_{\dot{f}}$.*

*Proof.* For simplicity, we consider the equivalent tax-free case that we have two costs $c, \dot{c}$ with subgradient lower bounded by $\epsilon$ and they only differ in facility $\dot{f}$. The potential functions are

$$\Phi(Y) = \sum_f \int_0^{Y_f} c_f(u) du, \ \dot{\Phi}(Y) = \sum_f \int_0^{Y_f} \dot{c}_f(u) du.$$

By Lemma 2, $\Phi$ and $\dot{\Phi}$ are strongly convex and thus the Nash equilibrium load $y$ and $\dot{y}$ are unique. Suppose $y_{\dot{f}} = \dot{y}_{\dot{f}}$. Consider any $Y \in \mathcal{Y}$ such that $Y_{\dot{f}} = y_{\dot{f}}$, we have

$$\dot{\Phi}(Y) - \dot{\Phi}(y) = \sum_f \int_{y_f}^{Y_f} \dot{c}_f(u) du = \sum_{f \neq \dot{f}} \int_{y_f}^{Y_f} \dot{c}_f(u) du + \int_{y_{\dot{f}}}^{Y_f} \dot{c}_f(u) du$$

$$= \sum_{f \neq \dot{f}} \int_{y_f}^{Y_f} c_f(u) du = \Phi(Y) - \Phi(y) \geq 0.$$

As a result, we have $\dot\Phi(y) \le \dot\Phi(\dot y)$. By the optimality of $\dot y$, we have $y = \dot y$. By contradiction, if $y \ne \dot y$, we have $y_{\hat f} = \dot y_{\hat f}$. $\qquad\square$

**Lemma 6.** *If $|u_1 - u_2| \le \Delta$, then for any $|u_3 - u_1| \le \Delta$, we have*

$$\left| \frac{c_f(u_1) - c_f(u_2)}{u_1 - u_2} - c'_f(u_3) \right| \le \epsilon.$$

*Proof.* This is a direct corollary of the $\beta$-smoothness. By mean value theorem, we have $\frac{c_f(u_1) - c_f(u_2)}{u_1 - u_2} = c'_f(u)$ for some $u \in [u_1, u_2]$. As $|u - u_3| \le |u - u_1| + |u_1 - u_3| \le 2\Delta \le \frac{\epsilon}{2\beta}$, we have

$$\left| \frac{c_f(u_1) - c_f(u_2)}{u_1 - u_2} - c'_f(u_3) \right| \le \epsilon.$$

$\qquad\square$

**Lemma 7.** *For round $t$ and facility $f$, if $u \in \mathcal{K}^t_f$, then we have $\left| \tau^t_f(u) - \tau^*_f(u) \right| \le 2\epsilon$.*

*Proof.* By the algorithm design, for each $u \in \mathcal{K}^t_f$, $\tau^t_f(u)$ will not change after $u$ is added to $\mathcal{K}_f$. We will use induction on $t$ to prove $\left| \hat\tau^t_f(u) - \tau_f(u) \right| \le \epsilon$ for $u \in \mathcal{K}^t_f$. At round $t = 1$, $\mathcal{K}^1_f = \{0\}$ and $\hat\tau^1_f(0) = \tau^*_f(0) = 0$ holds.

Suppose at round $t$, we have $\mathcal{K}^{t+1}_{\widetilde f^t} = \mathcal{K}^t_{\widetilde f^t} \bigcup \{u\}$ with $u \in \{[y^t_{\widetilde f^t}]^+_{\mathcal{L}}, [y^t_{\widetilde f^t}]^-_{\mathcal{L}}\} \backslash \mathcal{K}^t_{\widetilde f^t}$, and $\mathcal{K}^{t+1}_f = \mathcal{K}^t_f$ for $f \ne \widetilde f^t$. By the induction hypothesis, we only need to prove $\left| \hat\tau^{t+1}_{\widetilde f^t}(u) - \tau^*_{\widetilde f^t}(u) \right| \le 2\epsilon$. Recall that

$$\hat\tau^{t+1}_{\widetilde f^t}(u) = \mathrm{clip}\Big(u \cdot \frac{c^t_{\widetilde f^t} - \dot c^t_{\widetilde f^t}}{y^t_{\widetilde f^t} - \dot y^t_{\widetilde f^t}}, \hat\tau^t_{\widetilde f^t}([y^t_{\widetilde f^t}]^-_{\mathcal{K}_{\widetilde f^t}}), \hat\tau^t_{\widetilde f^t}([y^t_{\widetilde f^t}]^+_{\mathcal{K}_{\widetilde f^t} \bigcup\{1\}})\Big).$$

Then we have the following three cases. For simplicity we replace $\widetilde f^t$ with $f$.

(1) $\hat\tau^{t+1}_f(u) = u \cdot \frac{c^t_f - \dot c^t_f}{y^t_f - \dot y^t_f}$. By Lemma 11 and Lemma 6, we have

$$\left| \hat\tau^{t+1}_f(u) - \tau^*_f(u) \right| = \left| u \frac{c^t_f - \dot c^t_f}{y^t_f - \dot y^t_f} - u c'_f(u) \right| \le \epsilon.$$

(2) $\hat\tau^{t+1}_f(u) = \hat\tau^t_f([y^t_f]^-_{\mathcal{K}^t_f})$ and $u \cdot \frac{c^t_f - \dot c^t_f}{y^t_f - \dot y^t_f} \le \hat\tau^t_f([y^t_f]^-_{\mathcal{K}^t_f})$. Then we have

$$\hat\tau^{t+1}_f(u) = \hat\tau^t_f([y^t_f]^-_{\mathcal{K}^t_f}) \le \tau^*_f([y^t_f]^-_{\mathcal{K}^t_f}) + \epsilon \le \tau^*_f(u) + \epsilon,$$

where the first inequality is from the induction hypothesis as $[y^t_f]^-_{\mathcal{K}^t_f} \in \mathcal{K}^t_f$ and the second inequality is from Assumption 2. In addition, we have

$$\hat\tau^{t+1}_f(u) \ge u \cdot \frac{c^t_f - \dot c^t_f}{y^t_f - \dot y^t_f} \ge u c'_f(u) - \epsilon = \tau^*_f(u) - \epsilon.$$

(3) $\hat\tau^{t+1}_f(u) = \hat\tau^t_f([y^t_f]^+_{\mathcal{K}^t_f \bigcup\{1\}})$ and $u \cdot \frac{c^t_f - \dot c^t_f}{y^t_f - \dot y^t_f} \ge \hat\tau^t_f([y^t_f]^+_{\mathcal{K}^t_f \bigcup\{1\}})$. Then we have

$$\hat\tau^{t+1}_f(u) \le u \frac{c^t_f - \dot c^t_f}{y^t_f - \dot y^t_f} \le u c'_f(u) + \epsilon \le \tau^*_f(u) + \epsilon.$$

If $[y^t_f]^+_{\mathcal{K}^t_f \bigcup\{1\}} \in \mathcal{K}^t_f$, then we have

$$\hat\tau^t_f(u) = \hat\tau^t_f([y^t_f]^+_{\mathcal{K}^t_f \bigcup\{1\}}) \ge \tau^*_f([y^t_f]^+_{\mathcal{K}^t_f \bigcup\{1\}}) - \epsilon \ge \tau^*_f(u) - \epsilon.$$

If $[y_f^t]_{\mathcal{K}_f^t \bigcup \{1\}}^+ = 1$ and $1 \notin \mathcal{K}_f^t$, we still have

$$\widehat{\tau}_f^t(u) = \beta \geq uc_f'(u) = \tau_f^*(u).$$

For each of these three cases, the induction holds.

As $\tau_f^t(u) = \widehat{\tau}_f^t(u) + \epsilon u$ for all $f \in \mathcal{F}$ and $u \in [0,1]$, we have $\left|\tau_f^t(u) - \tau_f^*(u)\right| \leq 2\epsilon$. $\qquad\square$

**Lemma 8.** *For round $t$, if facility $f$ is known, then we have $\left|\tau_f^t(y_f^t) - \tau_f^*(y_f^t)\right| \leq 3\epsilon$.*

*Proof.* If $y_f^t \in \mathcal{K}_f^t$, we can directly apply Lemma 7. Otherwise, we set $u_1 = [y_f^t]_{\mathcal{L}}^-$ and $u_2 = [y_f^t]_{\mathcal{L}}^+$. Then we have $u_1 < y_f^t < u_2$ and $u_1, u_2 \in \mathcal{K}_f^t$. There exists $\lambda_1 \in [0,1], \lambda_1 + \lambda_2 = 1$ such that $y_f^t = \lambda_1 u_1 + \lambda_2 u_2$. By Lemma 6, we have $\left|\tau_f^t(u_i) - \tau_f^*(u_i)\right| \leq 2\epsilon$ for $i \in \{1,2\}$. Then we have

$$\left|\tau_f^t(y_f^t) - \tau_f(y_f^t)\right|$$
$$= \left|\lambda_1 \tau_f^t(u_1) + \lambda_2 \tau_f^t(u_2) - (\lambda_1 u_1 + \lambda_2 u_2)c_f'(u)\right|$$
$$\leq \left|\lambda_1 \tau_f^t(u_1) - \lambda_1 u_1 c_f'(u)\right| + \left|\lambda_2 \tau_f^t(u_2) - \lambda_2 u_2 c_f'(u)\right|$$
$$\leq \lambda_1 \left|\tau_f^t(u_1) - \tau_f^*(u_1)\right| + \lambda_1 u_1 \left|c_f'(u_1) - c_f'(u)\right| + \lambda_2 \left|\tau_f^t(u_2) - \tau_f^*(u_2)\right| + \lambda_2 u_2 \left|c_f'(u_2) - c_f'(u)\right|$$
$$\leq 2\lambda_1\epsilon + \lambda_1\epsilon + 2\lambda_2\epsilon + \lambda_2\epsilon \qquad\qquad \text{(Lemma 6, $\beta$-smoothness and $|u - u_i| \leq \epsilon/\beta$.)}$$
$$\leq 3\epsilon.$$

$\qquad\square$

**Lemma 9.** *If Algorithm 2 return* False *at round $t$, then for any $\widetilde{\tau} \in \mathbb{R}^F$ such that $\widetilde{\tau}_f = \tau_f^t(y_f)$ for $f \in \mathcal{F} \backslash \bar{\mathcal{F}}^t$ and $\widetilde{\tau}_f \in [l_f^t, r_f^t]$ for $f \in \bar{\mathcal{F}}^t$, we have $\mathrm{Gap}_i(x^t, c^t + \widetilde{\tau}) \geq 0$ for all $i \in [m]$. In addition, $x^t$ is a Nash equilibrium for tax $\widetilde{\tau}$.*

*Proof.* For simplicity, we will omit $t$ when there is no confusion. Algorithm 2 return False if and only if for all $i \in [m]$ and tax $\bar{\tau}_i : \bar{\tau}_{\bar{F}_i} = r_{\bar{F}_i}, \bar{\tau}_{\bar{F}_i'} = l_{\bar{F}_i'}, \bar{\tau}_{F \backslash (\bar{F}_i \bigcup \bar{F}_i')} = \tau_{F \backslash (\bar{F}_i \bigcup \bar{F}_i')}$, we have

$$\mathrm{Gap}_i(x, c + \bar{\tau}) = \min_{a:x_{i,a}=0} \sum_{f:f \in a} (c_f + \bar{\tau}_f) - \max_{a:x_{i,a} \neq 0} \sum_{f:f \in a} (c_f + \bar{\tau}_f) \geq 0.$$

By the definition of $\bar{F}_i$, for any $f \in \bar{F}_i$ and $a : x_{i,a} \neq 0$, we have $f \in a$. Similarly, for any $f \in \bar{F}_i'$ and $a : x_{i,a} \neq 0$, we have $f \notin a$. Thus for any $a : x_{i,a} \neq 0$, we have

$$\sum_{f:f \in a} (c_f + \bar{\tau}_f) - \sum_{f:f \in a} (c_f + \widetilde{\tau}_f) = \sum_{f \in \bar{F}_i} (r_f - \widetilde{\tau}_f) \geq 0.$$

For any $a : x_{i,a} = 0$, we have

$$\sum_{f:f \in a} (c_f + \bar{\tau}_f) - \sum_{f:f \in a} (c_f + \widetilde{\tau}_f) = \sum_{f \in \bar{F}_i' \bigcap a} (l_f - \widetilde{\tau}_f) \leq 0.$$

As a result, we have

$$\mathrm{Gap}_i(x^t, c^t + \widetilde{\tau}) = \min_{a:x_{i,a}=0} \sum_{f:f \in a} (c_f + \widetilde{\tau}_f) - \max_{a:x_{i,a} \neq 0} \sum_{f:f \in a} (c_f + \widetilde{\tau}_f)$$
$$\geq \min_{a:x_{i,a}=0} \sum_{f:f \in a} (c_f + \bar{\tau}_f) - \max_{a:x_{i,a} \neq 0} \sum_{f:f \in a} (c_f + \bar{\tau}_f) \geq 0.$$

To prove that $x^t$ is Nash equilibrium for tax $\widetilde{\tau}$, we only need to show that for in-support actions $a : x_{i,a}^t \neq 0$, the action costs $\sum_{f:f \in a} (c_f^t + \widetilde{\tau}_f)$ are the same. This can be derived by

$$\sum_{f:f \in a} (c_f^t + \tau_f^t) - \sum_{f:f \in a} (c_f^t + \widetilde{\tau}_f) = \sum_{f \in \bar{F}_i} (\tau_f^t - \widetilde{\tau}_f), \forall a : x_{i,a}^t \neq 0,$$

which is independent of $a$. As $x^t$ is Nash equilibrium for tax $\tau^t$, $\sum_{f:f \in a}(c_f^t + \tau_f^t)$ is also independent of $a$. $\qquad\square$

**Lemma 10.** *If Algorithm 2 return* False *at round t, then tax $\tau^t$ is an $6F\epsilon$-optimal tax.*

*Proof.* For known facility $f$, by Lemma 8, we have $\left|\tau_f^*(y_f^t) - \tau_f^t(y_f^t)\right| \le 3\epsilon$. By Lemma 7, for any $u \in \mathcal{K}_f^t$, we have $\left|\tau_f^*(u) - \tau_f^t(u)\right| \le 2\epsilon$. Thus for unknown facility $f$, we have

$$l_f^t = \tau_f^t([y_f^t]_{\mathcal{K}_f^t}^-) + \epsilon \cdot (y_f^t - [y_f^t]_{\mathcal{K}_f^t}^-) \le \tau_f^*([y_f^t]_{\mathcal{K}_f^t}^-) + 2\epsilon + \epsilon = \tau_f^*([y_f^t]_{\mathcal{K}_f^t}^-) + 3\epsilon,$$

$$r_f^t = \tau_f^t([y_f^t]_{\mathcal{K}_f^t \bigcup \{1\}}^+) + \epsilon \cdot (y_f^t - [y_f^t]_{\mathcal{K}_f^t \bigcup \{1\}}^+) \ge \tau_f^*([y_f^t]_{\mathcal{K}_f^t \bigcup \{1\}}^+) - 2\epsilon - \epsilon = \tau_f^*([y_f^t]_{\mathcal{K}_f^t \bigcup \{1\}}^+) - 3\epsilon,$$

As $\tau_f^*$ is nondecreasing (Assumption 2), we have

$$l_f^t - 3\epsilon \le \tau_f^*([y_f^t]_{\mathcal{K}_f^t}^-) \le \tau_f^*(y_f^t) \le \tau_f^*([y_f^t]_{\mathcal{K}_f^t \bigcup \{1\}}^+) \le r_f^t + 3\epsilon.$$

Thus there exists tax $\widetilde{\tau}^t$ such that $\widetilde{\tau}_f^t(y_f^t)$ satisfies the condition of Lemma 9 and $\left|\tau_f^*(y_f^t) - \widetilde{\tau}_f^t(y_f^t)\right| \le 3\epsilon$ for all $f \in \mathcal{F}$. $x^t$ is the Nash equilibrium for tax $\widetilde{\tau}^t$, we have

$$\forall i \in [m], a, a' \in \mathcal{A}_i, \sum_{f \in a} c_f(y_f^t) + \widetilde{\tau}_f^t(y_f^t) \le \sum_{f \in a'} c_f(y_f^t) + \widetilde{\tau}_f^t(y_f^t), \text{ if } x_{i,a}^t > 0.$$

Thus we have

$$\forall i \in [m], a, a' \in \mathcal{A}_i, \sum_{f \in a} c_f(y_f^t) + \tau_f^*(y_f^t) \le \sum_{f \in a'} c_f(y_f^t) + \tau_f^*(y_f^t) + 6F\epsilon, \text{ if } x_{i,a}^t > 0.$$

By Lemma 4, $\Psi(y_f^t) - \min_{y \in \mathcal{Y}} \Psi(y) \le 6F\epsilon$. $\qquad \square$

**Lemma 11.** *If Algorithm 2 output $\widetilde{\tau}^t, \widetilde{f}^t, \text{sign}^t$ at round t, then we have*

$$0 < \left|y_{\widetilde{f}^t}^t - \dot{y}_{\widetilde{f}^t}^t\right| \le \Delta.$$

*Proof.* First, we prove $\left|y_{\widetilde{f}^t}^t - \dot{y}_{\widetilde{f}^t}^t\right| > 0$. We consider the following two cases.

(1) Algorithm 2 return at Line 5. As we have $0 < y_i(\widetilde{f}^t) < w_i$, there exists $a, a' \in \mathcal{A}_i$ such that $x_{i,a} > 0, x_{i,a'} > 0$ and $\widetilde{f}^t \in a, \widetilde{f}^t \notin a'$. Suppose $y_{\widetilde{f}^t}^t = \dot{y}_{\widetilde{f}^t}^t$. Then by Lemma 3, we have $y^t = \dot{y}^t$ as $\tau^t$ and $\dot{\tau}^t$ only differ in facility $\widetilde{f}^t$. As a result, $x^t$ is Nash equilibrium for tax $\dot{\tau}^t$. However, $x^t$ is the Nash equilibrium for tax $\tau_f^t$ implies

$$\sum_{f \in a} c_f(y_f^t) + \tau_f^t(y_f^t) = \sum_{f \in a'} c_f(y_f^t) + \tau_f^t(y_f^t).$$

As $\tau^t$ and $\dot{\tau}^t$ only differ in facility $\widetilde{f}^t$ and $\widetilde{f}^t \in a, \widetilde{f}^t \notin a'$, we have

$$\sum_{f \in a} c_f(y_f^t) + \dot{\tau}_f^t(y_f^t) \ne \sum_{f \in a'} c_f(y_f^t) + \dot{\tau}_f^t(y_f^t),$$

which means $x^t$ is not the Nash equilibrium for tax $\dot{\tau}$. By contradiction, we have $y_{\widetilde{f}^t}^t = \dot{y}_{\widetilde{f}^t}^t$.

(2) Algorithm 2 return $\widetilde{\tau}^u, \widetilde{f}, \text{sign}$ at Line 23 or Line 31. As there exists $j \in [m]$ such that $\text{Gap}_j(x, c + \widetilde{\tau}^{u+\text{sign}\cdot\epsilon}) < 0$, $x$ is not a Nash equilibrium under tax $\dot{\tau}^t$. Let $\ddot{\tau}^t : \ddot{\tau}_f^t = \tau_f^t \bigcup (y_f^t, \widetilde{\tau}_f^u)$ for $f \in \mathcal{F}$. Then $\dot{\tau}^t$ and $\ddot{\tau}^t$ only differs in $\widetilde{f}$ and $x$ is the Nash equilibrium under tax $\ddot{\tau}^t$. By applying Lemma 3 with $\dot{\tau}^t$ and $\ddot{\tau}^u$, we have $y_{\widetilde{f}^t}^t = \dot{y}_{\widetilde{f}^t}^t$.

Second, we prove $\left|y_{\widetilde{f}^t}^t - \dot{y}_{\widetilde{f}^t}^t\right| \le \Delta$. (1) Algorithm 2 return at Line 5. Suppose we have $\left|y_f^t - \dot{y}_f^t\right| > \Delta$. By the tax design, the (sub)gradient of the tax $(\tau_f^t)'(u) \ge \epsilon$ for $u \in [0, 1]$. As a result, $\Phi(y, c + \tau^t)$ is $\epsilon$-strongly convex by Lemma 2. As $y^t = \text{argmin}_{y \in \mathcal{Y}} \Phi(y; c + \tau^t)$, we have

$$\Phi(\dot{y}^t; c + \tau^t) - \Phi(y^t; c + \tau^t) > \epsilon\Delta^2/2.$$

However, we have $|\Phi(y; c + \tau^t) - \Phi(y; c + \dot{\tau}^t)| \leq \delta$ for all $y \in \mathcal{Y}$. Thus we have

$$\Phi(\dot{y}^t; c + \dot{\tau}^t) - \delta \leq \Phi(y^t; c + \dot{\tau}^t) - \delta/2 \leq \Phi(y^t; c + \tau^t) \leq \Phi(\dot{y}^t; c + \tau^t) \leq \Phi(\dot{y}^t; c + \dot{\tau}^t) + \delta.$$

Comparing to the inequality above, we have $2\delta > \epsilon\Delta^2/2$, which is incorrect by the definition of $\delta$. By contradiction, we have $\left|y_f^t - \dot{y}_f^t\right| \leq \Delta$.

(2) Algorithm 2 return $\tilde{\tau}^u, \tilde{f}$, sign at Line 23 or Line 31. Let $\ddot{\tau}^t : \ddot{\tau}_f^t = \tau_f^t \bigcup(y_f^t, \tilde{\tau}_f^u)$ for $f \in \mathcal{F}$. Then $x$ is the Nash equilibrium under tax $\ddot{\tau}^t$. Let $\ddot{\tau} : \ddot{\tau}_f^t = \tau_f^t \bigcup(y_f^t, \tilde{\tau}_f)$ for all $f \in \mathcal{F}$. By the definition of $\ddot{\tau}^t$ and the feasible range $\tilde{\tau}_f \in [l_f, r_f]$, the subgradient of $\ddot{\tau}_f^t$ is lower bounded by $\epsilon$. As a result, $\Phi(\cdot; c + \ddot{\tau}^t)$ is $\epsilon$-strongly convex on $[0, 1]$. We can prove $\left|y_f^t - \dot{y}_f^t\right| \leq \Delta$ by following the analysis for case (1) and replacing $\tau^t$ with $\ddot{\tau}^t$.

$\square$

**Proposition 1.** *If Algorithm 2 return* False *at round t, then tax $\tau^t$ is an $6\epsilon F$-optimal tax. If Algorithm 2 output $\tilde{\tau}^t, \tilde{f}^t, \mathrm{sign}^t$ at round t, then we have*

$$0 < \left|y_{\tilde{f}^t}^t - \dot{y}_{\tilde{f}^t}^t\right| \leq \Delta.$$

*Proof.* This is directly from Lemma 10 and Lemma 11. $\square$

**Lemma 12.** *Algorithm 1 return* False *in at most $KF$ rounds.*

*Proof.* By Lemma 11 and the update rule (1), if Algorithm 2 return $\tilde{\tau}^t, f,$ sign at round $t$, then we will have one more known point, i.e., $\sum_{f \in \mathcal{F}} \mathcal{K}_f^{t+1} = \sum_{f \in \mathcal{F}} \mathcal{K}_f^t + 1$. As $\mathcal{K}_f^t \subseteq \mathcal{L}$ for all $f \in \mathcal{F}$ and $|\mathcal{L}| = K + 1$, we proved the lemma.

$\square$

**Theorem 1.** *Under Assumption 1 and Assumption 2, Algorithm 1 will output a $6\epsilon F$ tax within $T \leq 2F\beta/\epsilon$ rounds. In addition, each round has at most two tax realizations.*

*Proof.* The proof is directly from Proposition 1 and Lemma 12. $\square$

## C  Computation Complexity

In this section, we discuss the computation complexity of Algorithm 1 and Algorithm 2. We will show that these two algorithms can be implemented with $\widetilde{O}(\mathrm{poly}(A, F, m))$ complexity for each round. For network congestion games, the computation complexity can be sharpened to $\widetilde{O}(\mathrm{poly}(V, E, m))$, avoiding the dependence on $A$ that can be exponential in $V$ and $E$.

### C.1  General Congestion Games

For Algorithm 1, we compute/update the value of the cost/tax function for each facility. As we use the dictionary data structure, computing value and updating value only have $O(\log K) = O(\log \beta/\epsilon)$ complexity. As a result, the complexity of one round in Algorithm 1 is $\widetilde{O}(F)$.

For Algorithm 2, $x \in \phi^{-1}(y)$ is a Caratheodory decomposition problem and can be formulated as a linear program with $A$ variables, $F + m$ equation constraints and $A$ inequality constraints (Proposition 2), which can be solved in polynomial time [Cohen et al., 2021].

The bottleneck is in computing $u = \mathrm{argmax}_u \{u : \mathrm{Gap}_j(x, c + \tilde{\tau}^u) \geq 0, \forall j \in [m]\}$ for $\tilde{\tau}^u : \tilde{\tau}_f^u = u, \tilde{\tau}_{\mathcal{F}\backslash\{f\}}^u = \tau'_{\mathcal{F}\backslash\{f\}}, f \in \bar{F}_i$. For simplicity, we use the notation: $\tilde{c}^u = c + \tilde{\tau}^u$ as the cost with tax $\tilde{\tau}^u$. By Definition 5 and the definition of action cost, we have

$$\mathrm{Gap}_j(x, c + \tilde{\tau}^u) = \min_{a:x_{j,a}=0} \tilde{c}_a^u - \max_{a:x_{j,a}\neq 0} \tilde{c}_a^u. \tag{4}$$

For action cost $\widetilde{c}_a^u$, if $f \in a$, it is a linear function w.r.t. $u$ in the form of $u + C$ for some constant $C$. Otherwise, it is a constant w.r.t. $u$. As a result, we can determine the function $\widetilde{c}_a^u$ with $O(F)$ computation as we only need to compute $\widetilde{c}_a^{\tau_f'}$ to decide the constant. Then we can compute $\mathrm{Gap}_j(x, c + \widetilde{\tau}^u)$ in closed form and compute $u_j = \mathrm{argmax}_u\{u : \mathrm{Gap}_j(x, c + \widetilde{\tau}^u) \geq 0\}$ with $O(AF)$ complexity. Finally, $u = \min_{j \in [m]} u_j$ can be computed with $\mathcal{O}(mAF)$ complexity. Similarly, $u = \mathrm{argmin}_u\{u : \mathrm{Gap}_j(x, c + \widetilde{\tau}^u) \geq 0, \forall j \in [m]\}$ has $\widetilde{O}(mAF)$ computation complexity.

## C.2  Network Congestion Games

For network congestion games, Algorithm 2 can be implemented by applying shortest path algorithms on a modified network, thus avoiding the dependence on $A$. We will apply Dijkstra's algorithm with $\widetilde{O}(V + E)$ complexity while other shortest path algorithms can be used as well.

First, the Caratheodory decomposition $x \in \phi^{-1}(y)$ can be done efficiently with $O(VE + E^2)$ steps similar to the decomposition algorithm in [Panageas et al., 2023]. While their algorithm is for the flow polytopes with one commodity, it can be directly generalized to the multi-commodity case. We defer the algorithm and analysis to Appendix D.

For $u = \mathrm{argmax}_u\{u : \mathrm{Gap}_j(x, c + \widetilde{\tau}^u) \geq 0, \forall j \in [m]\}$, the computation complexity can be boosted to $\widetilde{O}(m(E + V))$. To achieve this, we consider how (4) changes as $u$ increases from $\tau_f'$ to $r_f$. By Algorithm 2, we have $\mathrm{Gap}_j(x, c + \widetilde{\tau}^u) \geq 0$ when $u = \tau_f'$ as otherwise the algorithm ends at the previous iteration. In addition, facility $f$ either has none of the Nash load or has all of the Nash load for facility $j$ according to the algorithm design. For the first case, the in-support action costs will not change as $u$ increases. $\mathrm{Gap}_j(x, c + \widetilde{\tau}^u) \geq 0$ always holds as the off-support action costs are nondecreasing w.r.t. $u$.

For the second case (all in-support actions use $f$), the in-support action costs take the form of $u + C$ and $C$ can be determined by applying shortest path algorithm with edge weight $c + \widetilde{\tau}^{\tau_f'}$. For off-support action cost, we observe that

$$\min_{a : x_{j,a} = 0} \widetilde{c}_a^u = \min\Big\{ \min_{a : x_{j,a} = 0, f \in a} \widetilde{c}_a^u, \min_{a : x_{j,a} = 0, f \notin a} \widetilde{c}_a^u \Big\} = \min\Big\{ \min_{a : x_{j,a} = 0, f \in a} \widetilde{c}_a^u, \min_{a : f \notin a} \widetilde{c}_a^{\tau_f'} \Big\}, \quad (5)$$

where the second equation is from that the action cost does not depend on $u$ and $x_{j,a} = 0$ if $f \notin a$. The first term in (5) grows linearly w.r.t. $u$ as $\widetilde{f} \in a$, so it is always larger than the in-support action cost. The second term in (5) is the shortest path length for commodity $j$ that does not use facility $f$, which can be computed as the shortest path in the network after removing edge $f$. As a result, $u_j = \mathrm{argmax}_u\{u : \mathrm{Gap}_j(x, c + \widetilde{\tau}^u) \geq 0\}$ can be computed with $O(E + V)$ complexity. Then the complexity for computing $u = \min_{j \in [m]} u_j$ is $\widetilde{O}(m(E + V))$.

Similarly, $u_j = \mathrm{argmin}_u\{u : \mathrm{Gap}_j(x, c + \widetilde{\tau}^u) \geq 0\}$ can be reduced to solving the shortest path that must use edge $f$ in the network. We consider how (4) changes as $u$ decreases from $\tau_f$ to $l_f$. Initially, the gap is nonnegative. If $f$ has all of the Nash load, then the in-support action cost is a linear function $u + C$ and it decreases at least as fast as the first term. As a result, the gap is always nonnegative. Otherwise, $f$ has none of the Nash load and in-support action costs remain constant.

We notice the following equation:

$$\min_{a : x_{j,a} = 0} \widetilde{c}_a^u = \min\Big\{ \min_{a : x_{j,a} = 0, f \in a} \widetilde{c}_a^u, \min_{a : x_{j,a} = 0, f \notin a} \widetilde{c}_a^u \Big\} = \min\Big\{ \min_{a : f \in a} \widetilde{c}_a^u, \min_{a : x_{j,a} = 0, f \notin a} \widetilde{c}_a^{\tau_f'} \Big\}, \quad (6)$$

where the second equation is from that $f \in a$ implies $a$ is off-support ($x_{j,a} = 0$) and $f \notin a$ implies $\widetilde{c}_a^u$ is independent of $u$. The second term in (6) is a constant and is always greater than the in-support action cost. The first term in (6) is a linear function $u$ and it can be determined by computing the shortest path that always uses $\widetilde{f}$ and with edge weights $c + \widetilde{\tau}^{\tau_f}$. This subproblem can be solved by applying the shortest path algorithm twice: the first one is to connect the source node and the starting node of $\widetilde{f}$, and the second one is to connect the end node of $\widetilde{f}$ and the target node. As a result, the complexity for $u = \max_{j \in [m]} u_j$ is $\widetilde{O}(m(E + V))$ as well. Thus the computation complexity for Algorithm 2 in network congestion games is $O(VE + E^2 + mV + mE)$.

# D Missing Proofs in Section C

**Proposition 2.** *Finding $x \in \phi^{-1}(y)$ can be formulated as the following linear program.*

$$\min_{x \in \mathbb{R}^A} 1$$

$$s.t.\ y = \sum_{i \in [m]} \sum_{a_i \in \mathcal{A}_i} x_{i,a_i} a_i$$

$$w_i = \sum_{a_i \in \mathcal{A}_i} x_{i,a_i}, \forall i \in [m]$$

$$x_{i,a_i} \geq 0, \forall i \in [m], a_i \in \mathcal{A}_i$$

*Proof.* The second and third constraints guarantees $x$ is a feasible strategy. The first constraint indicates $y = \phi(x)$. As a result, any feasible point of the program is a solution of $\phi^{-1}(y)$. □

---

**Algorithm 3** Efficient Computation of Flow Decomposition (Modified from [Panageas et al., 2023])

1: **Input**: A load $y \in \mathcal{Y}$.
2: $x_{i,a} = 0$ for all $i \in [m]$ and $a \in \mathcal{A}_i$.
3: **while** $\exists f : y_f > 0$ **do**
4:     Let $A = \{f : y_f > 0\}$.
5:     Let $f_{\min} = \arg\min_{f \in A} y_f$ and $y_{\min} = \min_{f \in A} y_f$.
6:     Let $a$ be a $(s_i, t_i)$ path of network $G(V, A)$ with $f_{\min} \in p$.
7:     Let $x_{i,a} = y_{\min}, y_f = y_f - y_{\min}$ if $f \in a$.
8: **end while**

---

**Proposition 3.** *Algorithm 3 can output a Caratheodory decomposition of $y$ within $E$ steps.*

*Proof.* During the algorithm, load $y$ will always be nonnegative: $y_f \geq 0, \forall f \in \mathcal{F}$. For each round, we will have $y_{f_{min}}$ reduced to 0. As a result, the algorithm will end within at most $E$ rounds.

We only need to prove that path $a$ always exists in Line (6) for each round. First, $y$ always remains a multi-commodity flow as Line (7) will not affect the law of conservation in the network. By flow decomposition theorem, there exists simple paths $a_1, a_2, \cdots, a_p$ such that

$$y = \sum_{i \in [p]} w_i a_i,$$

where $w_i > 0$ are positive flow weights. As $y_{f_{\min}} > 0$, there exists $a_i$ such that $f_{\min} \in a_i$. Then for any $f \in a_i, y_f \geq w_i > 0$. As a result, path $a$ exists for Line (6). □

# E Experiments

We implemented our algorithm and conducted experiments on a classic example known as the nonlinear variant of Pigou's example [Nisan et al., 2007]. Concretely, nonlinear variant of Pigou's example is a routing game with one source node $s$ and one target node $t$. There are two edges connecting $s$ and $t$. One edge has constant cost $c_0(x) = c, \forall x \in [0, 1]$ for some $c \in [0, 1]$, and the other edge has polynomial cost $c_1(x) = x^p$. One important property of such games is the price of anarchy grows without bound as $p \to \infty$, which urges proper tax to induce socially optimal behavior.

We apply our algorithm to learn the optimal tax with different $c_0$ and $p$. As we can see, the social welfare quickly converges to the optimal one. Another important observation is the learned tax function does not uniformly converge to the marginal cost tax, which is reasonable as accurate estimate is only necessary around the Nash equilibrium induced by the tax.

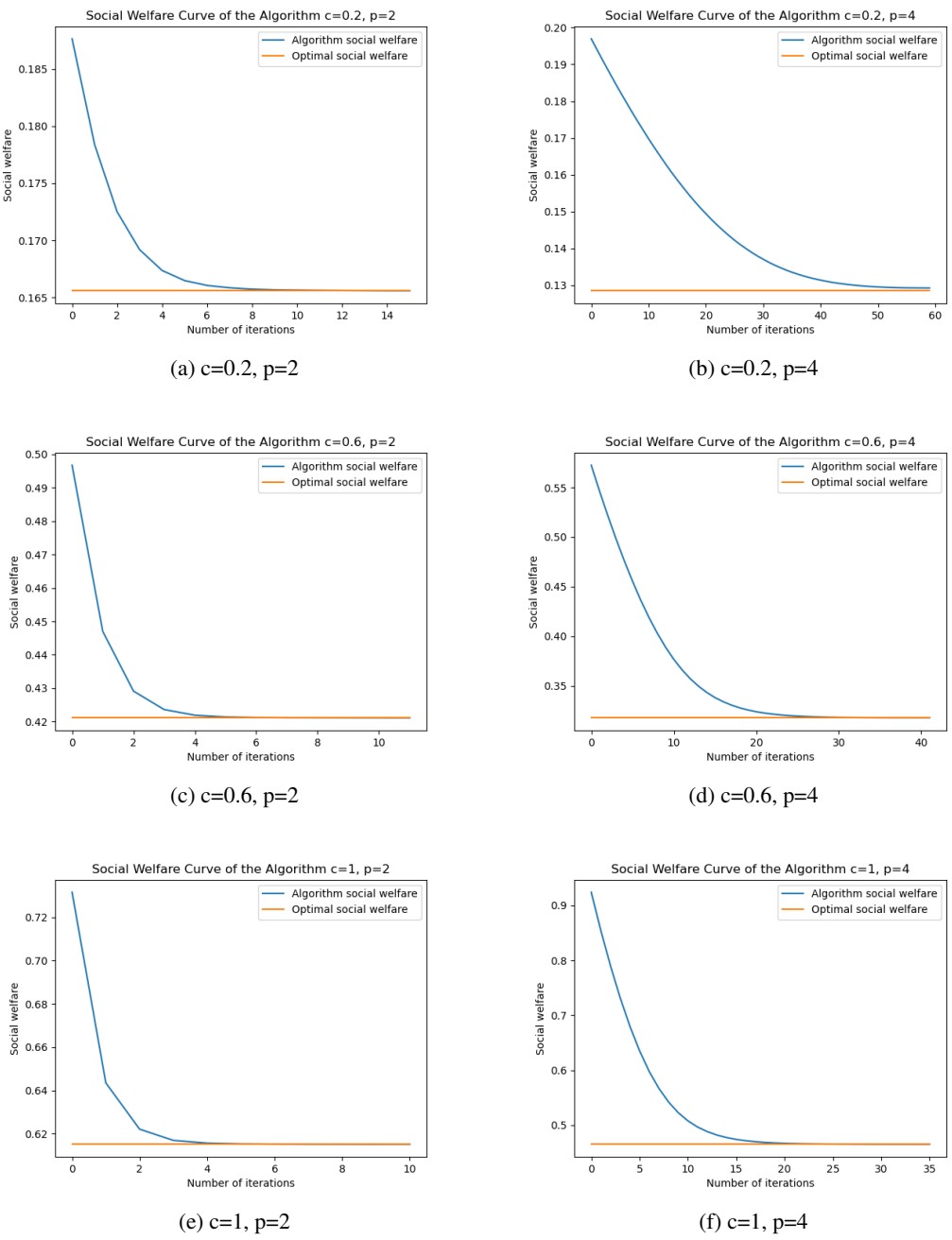

Figure 1: Social Welfare Curves of the Algorithm for various values of $c$ and $p$. We can observe that the social welfare converges to the optimal one quickly.

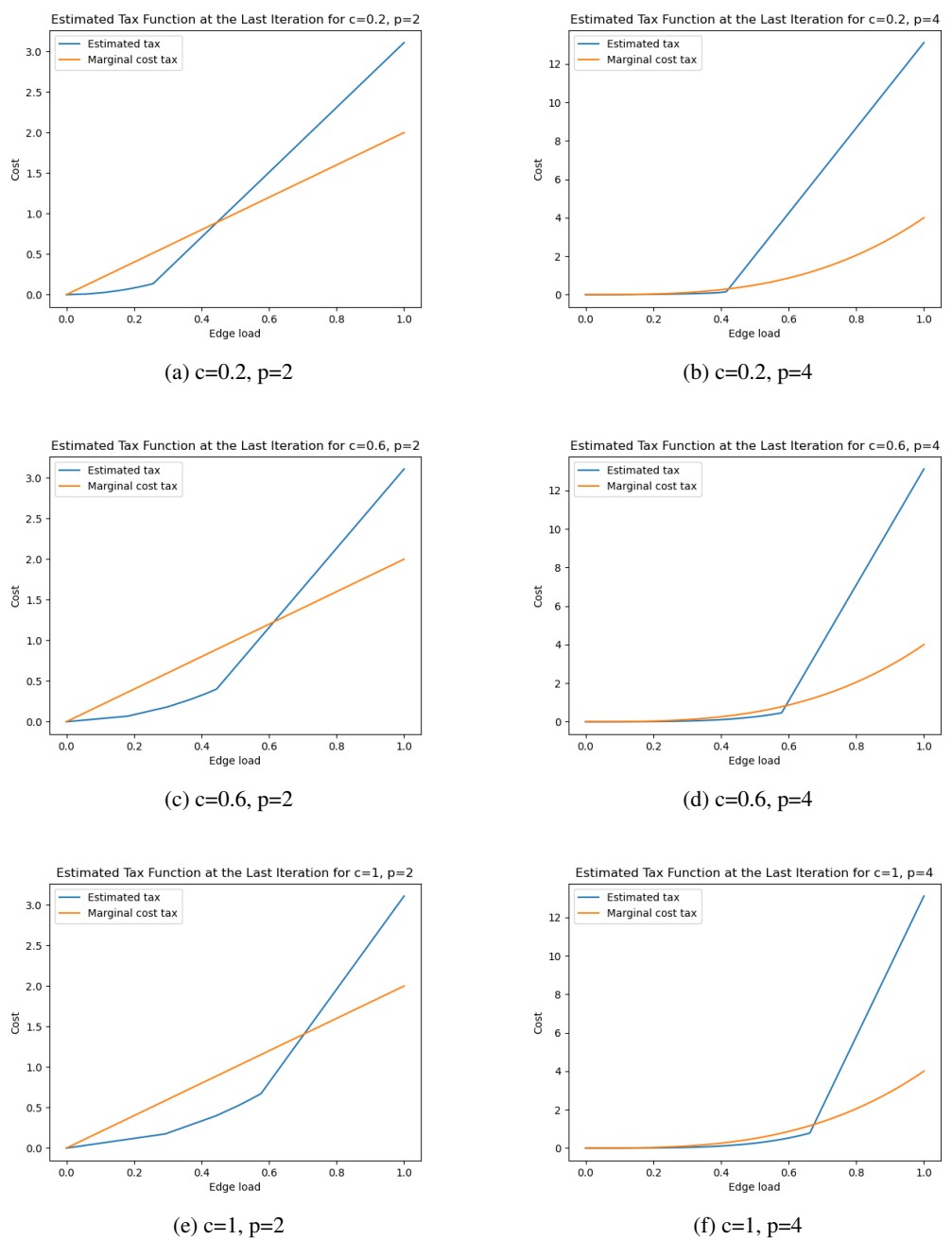

Figure 2: Estimated Tax Functions at the Last Iteration for various values of $c$ and $p$. The estimation is not uniformly accurate but they are accurate at the induced Nash equilibrium.

