# OpenReview forum: "Learning Optimal Tax Design in Nonatomic Congestion Games"
_NeurIPS.cc/2024/Conference — NeurIPS 2024 poster_

### Official Review · Reviewer_18G8 · 2024-07-06

**Soundness:** 3
**Presentation:** 2
**Contribution:** 4
**Rating:** 7
**Confidence:** 3

**Summary:**

This paper investigated how to design the optimal tax in congestion games to minimize the social cost, with partial information. Specifically, the tax designer cannot observe the cost function directly. Instead, whenever a tax assignment is announced, the designer will observe the Nash equilibrium and cost.

**Strengths:**

- This paper learns the optimal tax with partial information feedback, which is more realistic in real-world applications.
- The assumption is mild: This paper only assumes monotonicity and smoothness of the cost function, as well as a non-decreasing marginal cost tax
- The sample complexity is independent of the action set size

**Weaknesses:**

- There are some typos in the paper, for instance:
  - In definition of $\epsilon$-optimal tax, it should be $\Psi(y)\leq \min \Psi(y')$ instead of $\arg\min$
  - Line 163: Marginal cost tax
- The experiment is weak: The network has only two nodes and two edges.

**Questions:**

- What's $c'_f$ in the definition of marginal cost tax? Is it the derivative of the cost function?
- Need the algorithm to know the smoothness $\beta$ in advance?

**Limitations:**

Yes. The author addressed the assumptions required by the paper.

---

> ### Author Rebuttal · Authors · 2024-08-07
>
> Thanks for your appreciation! We address your questions below.
>
> - **Typos:** Thanks for pointing out the typos! We will correct them in the final version.
> - **Definition of the marginal cost function $c'_f$:** Yes it is the derivative of the cost function.
> - **Knowledge of $\beta$:** The algorithm requires knowing the value (or an upper bound) of the smoothness $\beta$ in advance. It would be an interesting future direction to remove this requirement.

---

> > ### Comment · Reviewer_18G8 · 2024-08-10
> >
> > Thank you for your response and I will keep my rating.

---

### Official Review · Reviewer_udJm · 2024-07-06

**Soundness:** 3
**Presentation:** 3
**Contribution:** 4
**Rating:** 7
**Confidence:** 3

**Summary:**

This manuscript proposes an algorithm for obtaining optimal tax design on nonatomic congestion games that decreases the social welfare of equilibrium state. Although the optimal tax design for nonatomic congestion games has a closed form of cost function, the cost function is generally unknown for tax designers. Thus, this manuscript proposes a non-parametric way to obtain an $\epsilon$-optimal tax design under that the tax designer can only access the information of equilibrium state. The algorithms for network congestion games where each strategy is a path in the network is also given, which enjoy polynomial complexity with respect to the number of vertices and edges.

**Strengths:**

The proposed approach consists of novel and non-trivial ingredients: parameter-free approach with piecewise-linear tax function and tax design leading to strongly convex potential. Although there are some works for mathematical programming under equilibrium constraint (MPEC) that is a more generalized problem, they cannot reach a global optimal for tax design due to the non-convexity.

The support for network congestion games is also impressive, since it have exponentially many choices (paths) for users.

**Weaknesses:**

- The applicability of the proposed algorithm for network congestion games is not empirically confirmed. Since the proposed algorithm for network congestion games enjoy polynomial complexity, empirical evaluation with network congestion game instances will enhance the manuscript since it is directly connected to toll design on road networks.

Minor comments:
- Although I agree that it may be the first work to design tax under "equilibrium feedback", the notion of such a partial feedback setting may go back to previous literatures, e.g., [Liu et al., NeurIPS 2022] that is cited by the manuscript. Thus, it is questionable that the "equilibrium feedback" setting is the proposal of this manuscript.
- The limitations of the proposed approach should be summarized at the end of article.

**Questions:**

I'm interested in a variant of network congestion games, where user's strategy forms a specific type of subgraph (other than path) of a network. If the type has a polynomial time algorithm for optimizing the weight (e.g., Kruskal algorithm for spanning trees), can you incorporate this algorithm to the proposed approach? This may widen the applicability of the proposed algorithm with exponentially many strategies.

**Limitations:**

The limitations of the proposed approach should be summarized at the end of article.

---

> ### Author Rebuttal · Authors · 2024-08-07
>
> Thanks for your appreciation! We address your questions below.
>
> - **Equilibrium feedback:** Thanks for pointing out the usage of similar concepts as equilibrium feedback in prior works. We will comment on this and make clear that we are the first studying it in the tax design setting in the final version.
> - **Limitation:** We will add a limitation section at the end of the paper in the final version to summarize the limitations we mentioned in the main paper.
> - **Variant of network congestion games:** This is an very interesting direction! In our analysis, we only require that there are polynomial algorithms to optimize the weight in two specific networks, i.e., removing one given edge from the original network or having to use one given edge in the original network. For the minimum spanning tree case you mentioned, we can apply any minimum spanning tree with the weight of that given edge modified to be negative infinity or positive infinity (or just $-\sum_{e}w_e$ and $\sum_{e}w_e$).

---

> > ### Comment · Reviewer_udJm · 2024-08-13
> >
> > Thank you for detailed reply. The applicability for variants of network congestion games seems reasonable and interesting. Since I think this work is both theoretically and empirically important for tax design, I will keep my score.

---

### Official Review · Reviewer_vts3 · 2024-07-09

**Soundness:** 3
**Presentation:** 3
**Contribution:** 3
**Rating:** 7
**Confidence:** 4

**Summary:**

The paper studied the classic congestion pricing problem, which can be framed as a Stackelberg game in which a leader (a tax designer) imposes congestion tolls in a congested network used by many self-interested travelers (followers), with the goal of minimizing the total social cost.  The proposed algorithm can be applied to scenarios where the tex designer can observe nothing but the equilibrium reached by the travelers.

**Strengths:**

In the literature, most algorithms for congestion pricing are dedicated to scenarios where the tax designer can accurately predict how travelers would behave in response to the congestion taxes. However, for this to be possible, a basic requirement is that the tax designer should know how travelers would balance between time and money (like "$5" and "10 min", which one is more important), which is usually not practical. Hence, exploring congestion pricing under the bandit setting is a great endeavor.

Meanwhile, I believe there is no easy path to this setting, as it is fundamentally different from standard bandit problems: in this case, the "feedback" is an equilibrium state. Hence, I personally believe the paper's contribution is valid.

**Weaknesses:**

1. The review of the literature can be more solid.

1.1. In Appendix A, it is stated that "MPEC is a bilevel optimization problem, which is usually intractable." This is not sufficient. Instead, the authors are encouraged to cite some works that have proved MPECs to be strongly NP-hard (you may find many). Particularly, you may cite some works discussing the complexity of congestion pricing problems; the following paper is an example.

[1] Harks, Tobias, et al. "Computing network tolls with support constraints." Networks 65.3 (2015): 262-285.


1.2. I encourage the authors to add a paragraph in Section 2 to discuss "Learning in Stackelberg games with bandit feedback."

1.3. There are some other recent works on congestion pricing, which is related tot the present study.

[1] Li, Jiayang, et al. "Achieving hierarchy-free approximation for bilevel programs with equilibrium constraints." International Conference on Machine Learning. PMLR, 2023.

[2] Grontas, Panagiotis D., et al. "BIG Hype: Best intervention in games via distributed hypergradient descent." IEEE Transactions on Automatic Control (2024).

[3] Maheshwari, Chinmay, et al. "Convergent first-order methods for bi-level optimization and stackelberg games." arXiv preprint arXiv:2302.01421 (2023).


2. There are many editorial mistakes in the paper. For example, please check the title of Section 5.

For other weaknesses, please refer to my questions.

**Questions:**

My rating of the paper could be further increased if the following two questions can be appropriately answered.

1. In this paper, travelers' behavior is assumed to be a Nash equilibrium,  which implicitly assumes travelers are perfectly rational. A more realistic assumption is that they are boundedly rational, which may correspond to a so-called quantal response equilibrium. Please discuss how your work can be extended with this assumption.

2. In reality, equilibrium-reaching may be time-consuming, maybe taking several weeks. That is to say, it is not easy for a tax designer to actually observe the equilibrium outcome. How would you address this challenge?

**Limitations:**

The limitations are appropriately discussed.

---

> ### Author Rebuttal · Authors · 2024-08-07
>
> Thanks for your appreciation! We address your questions below.
>
> - **Literature review:** Thanks for pointing out references! We will include MPEC hardness and the complexity of the pricing problems in the final version. We will also move the discussion on Stackelberg games in the Appendix to the main paper.
> - **Quantal response equilibrium:** We consider Nash equilibrium as we believe this would be a reasonable first step in this direction. Relaxing this assumption, e.g. cosidering quantal response equilibrium, would be an interesting future direction. The main challenge in adapting our algorithm would be in finding the "boundary" tax that can provide critical information about the game. Designing such tax would require a more sophisticated analysis that is specialized to quantal response equilibrium.
> - **Time constraint in practice:** Our algorithm can be easily adapted to the case where we have feedbacks other than only the equilibrium feedback, which correspond to the observations when the system is evolving towards the Nash equilibrium. Specifically, when the tax designer obtain a non-equilibrium feedback, she can still update the optimal tax estimate (if the feedback provides new information) and apply the new tax. It is possible for our algorithm to find the optimal tax even if no equilibrium feedback is used. In addition, as long as the equilibrium can be reached after applying a tax for some time, the algorithm can always find the optimal tax. In short, our algorithm is not restricted to only using the equilibrium feedback, but all possible feedback.

---

### Official Review · Reviewer_Q6MU · 2024-07-12

**Soundness:** 3
**Presentation:** 3
**Contribution:** 3
**Rating:** 6
**Confidence:** 4

**Summary:**

The paper considers the problem where a system designer endeavors to guide players to welfare-maximizing behavior via imposing a tax function. One key challenge is that the designer can only observe equilibrium feedback from a Nash equilibrium, a stable state of the system, after imposing the tax. The main contribution of the paper is an algorithm that solves this problem with a polynomial sample complexity.

**Strengths:**

The main problem introduced in the paper is interesting, well-motivated and, to my knowledge, new. Imposing taxes so as to mitigate the price of anarchy has been a popular approach in the literature on algorithmic game theory, but prior work makes assumptions that are too strong for some applications, such as precise knowledge of the underlying game. Instead, this paper puts forward a more realistic feedback model, which I believe is well-motivated. Further, the proposed algorithm and analysis contains a number of interesting technical ideas, as this problem has a number of distinct features compared to prior work. These ideas are nicely presented in the introduction. Another compelling aspect of the proposed method is that it is also computationally efficient, even in network congestion games where the number of actions is exponential. The paper also accurately places its contributions in the context of related lines of work.

**Weaknesses:**

One important question, which I feel is not addressed adequately in the paper, is whether it is realistic to assume that players will immediately reach a Nash equilibrium after the imposition of the tax. This is an important modeling assumption which requires further justification in my opinion. If I understand correctly, a Nash equilibrium can be at least computed efficiently even after the tax is imposed; can the authors discuss about specific dynamics that will end up in an equilibrium in polynomial time? Otherwise, why is it reasonable to assume that a Nash equilibrium will be reached in the first place? Also, have the authors considered relaxing that assumption by only assuming that players have no-regret? For example, this assumption is made in the paper by Zhang et al. (titled "Steering No-Regret Learners to a Desired Equilibrium"), which could be discussed in the paper.

Besides the point above, it can be argued that most of the technical components used in the paper are fairly standard, although the way they are combined appears to be novel and non-trivial.

Finally, I believe the writing is a bit sloppy at times. The abstract talks about "boundary tax," which is not at all clear in the context of the abstract, although it is clarified later. Can the authors try to rephrase that in the abstract? I would also suggest pointing out in the abstract that the proposed algorithm is also computationally efficient, as it is an important feature. Moreover, the section of the conclusions is underwhelming, and I would strongly recommend expanding it with directions for future work. The authors should also remove the instructions block from the NeurIPS paper checklist so as to adhere to the formatting instructions.

**Questions:**

See above.

**Limitations:**

The authors have adequately addressed the limitations.

---

> ### Author Rebuttal · Authors · 2024-08-07
>
> Thanks for your appreciation! We address your questions below.
>
> - **Nash equilibrium assumption:** Under Assumption 1 (standard assumption used in nonatomic congestion game [Nisan et al., 2007]), the potential function of the game is convex and computing the Nash equlibrium is equivalent to solving a convex optimization problem. In addition, as we always apply a tax with positive subgradient lower bound, the potential function in the game with tax is always strongly convex (Lemma 2). As a result, computing the Nash equilibrium is equivalent to minimizing a strongly convex function. E.g., one kind of no-regret dynamics is shown to converge to Nash equilibrium [Chen and Lu, 2016]. Only assuming each player apply an no-regret algorithm would be an interesting future direction. We will add the discussion on [Zhang et al., 2023] in the final version.
> - **Writing:** Thanks for your suggestions! We will add the computation efficiency part to the abstract and replace boundary tax with "exploratory tax that can provide critical information about the game". We will also add a future work section with following potential directions.
> 1. Relaxing the Nash equilibrium assumption to players following no-regret dynamics or quantal response equilibrium.
> 2. Design algorithms that do not require prior knowledge of the smoothess coefficient.
> 3. Generalize the algorithm to atomic congestion games.
>
> > Chen, Po-An, and Chi-Jen Lu. "Generalized mirror descents in congestion games." Artificial Intelligence 241 (2016): 217-243.

---

> > ### Comment · Reviewer_Q6MU · 2024-08-11
> >
> > I thank the authors for the response. I have no further questions.

---

### Official Review · Reviewer_nncK · 2024-07-13

**Soundness:** 2
**Presentation:** 3
**Contribution:** 3
**Rating:** 4
**Confidence:** 3

**Summary:**

The paper focuses on the challenge of designing tax mechanisms to maximize social welfare in congestion games, where players' self-interested behaviors can lead to suboptimal outcomes for the overall system. The study introduces an innovative algorithm to learn the optimal tax with limited feedback, termed "equilibrium feedback," where the tax designer can only observe the Nash equilibrium resulting from a deployed tax plan.
This paper introduces a novel approach to learning optimal tax design in nonatomic congestion games, addressing critical challenges with innovative solutions. The proposed algorithm not only advances the theoretical understanding of congestion game tax design but also has potential real-world applications in improving system efficiency in scenarios with shared resources.

**Strengths:**

1. The paper introduces a novel algorithm for learning optimal tax design in nonatomic congestion games with equilibrium feedback, an approach that has not been previously explored. The use of piece-wise linear functions to approximate optimal tax functions is a good solution to handle the complexity of the tax function space.
2. The theoretical foundation of the paper is strong, with rigorous proofs and a well-defined algorithm that addresses key challenges in tax design.
3. The paper is well-organized and clearly written, making it easy to follow the logical progression of ideas and the development of the algorithm.

**Weaknesses:**

1. While the approach is novel, it heavily relies on theoretical constructs without introducing substantial new empirical methodologies or experimental techniques.
2. The absence of empirical validation or simulation results weakens the overall quality of the paper, as it is difficult to assess the practical effectiveness and efficiency of the proposed algorithm.
3. The paper's focus on theoretical development without sufficient empirical support may reduce its immediate relevance and usefulness to practitioners in the field.

**Questions:**

1. Can you provide empirical validation or simulation results to support the theoretical claims?
2. Can you justify the assumptions made about the smoothness and other properties of cost functions in more detail?
3. Can you compare your approach with existing tax design methods in terms of performance and applicability?

---

> ### Author Rebuttal · Authors · 2024-08-07
>
> Thanks for your thoughtful review. We acknowledge the importance of empirical validation and **have provided it in Appendix F**. In the final version, we will ensure this is better highlighted.
>
> - **Assumptions:** Monotonicity is a standard assumption in the nonatomic congestion game literature [Nisan et al., 2007]. Smoothness is a widely accepted technical assumption in the literature. Non-decreasing marginal cost function is a reasonable assumption in real-world problems that enjoy the law of diminishing return. We want to emphasize that our assumptions are much weaker than all prior works, which assumes at least one of the following conditions: full information about the game, (linearly) parameterized cost function, control over all the agent, and convex objective function.
> - **Comparision with existing methods:** One line of research focused on the computational complexity of tax design in congestion game, which requires full knowledge of the game and is not applicable to the equilibrium feedback setting. Tax design with equilibrium feedback can also be formalized as Stackelberg games or mathematical programming under equilibrium constraint. However, existing approaches for these two problems require assumptions such as parameterized function space, convexity and control over the agents [Bai et al., 2021, Zhong et al., 2021, Li et al., 2020, Liu et al., 2022]. To best of our knowledge, none of the existing approaches has favorable theoretical guarantees. We believe our work serve as an initial step in this area.

---

### Decision · Program_Chairs · 2024-09-25

**Decision:**

Accept (poster)

**Comment:**

Based on the general consensus that this paper provides an important theoretical contribution, I recommend acceptance of the paper as a poster. Some major points of criticism were the lack / small scale of experiments and the practicability of the assumption that players converge towards an actual Nash equilibrium. However, the former is fine for a theoretical paper and the latter is indeed a limitation that should be clearly stated, but the results still remain interesting with this assumption.